# Implicit Preference Alignment for Human Image Animation

**Yuanzhi Wang** [*]   **Xuhua Ren   Jiaxiang Cheng   Bing Ma   Kai Yu**
**Tianxiang Zheng   Qinglin Lu   Zhen Cui**

## Abstract

Human image animation has witnessed significant advancements, yet generating high-fidelity hand motions remains a persistent challenge due to their high degrees of freedom and motion complexity. While reinforcement learning from human feedback, particularly direct preference optimization, offers a potential solution, it necessitates the construction of strict preference pairs. However, curating such pairs for dynamic hand regions is prohibitively expensive and often impractical due to frame-wise inconsistencies. In this paper, we propose *Implicit Preference Alignment (IPA)*, a data-efficient post-training framework that eliminates the need for paired preference data. Theoretically grounded in implicit reward maximization, IPA aligns the model by maximizing the likelihood of self-generated high-quality samples while penalizing deviations from the pretrained prior. Furthermore, we introduce a Hand-Aware Local Optimization mechanism to explicitly steer the alignment process toward hand regions. Experiments demonstrate that our method achieves effective preference optimization to enhance hand generation quality, while significantly lowering the barrier for constructing preference data. Codes are released at `https://github.com/mdswyz/IPA`

## 1. Introduction

Human image animation is a compelling yet challenging task, aiming to synthesize photorealistic videos that faithfully follow a reference image and a target pose sequence. This technology possesses significant transformative potential, with broad-reaching applications spanning filmmaking, advertising, and digital avatar synthesis (Cheng et al., 2025).

The field has witnessed a paradigm shift from early Generative Adversarial Networks (GANs)-based approaches (Li et al., 2019; Zhao & Zhang, 2022) to recent diffusion-based architectures (Hu, 2024; Zhang et al., 2025). Representative diffusion-based frameworks, such as Animate Anyone (Hu, 2024), introduced ReferenceNet to extract and align detailed appearance features for high-fidelity video generation. MimicMotion (Zhang et al., 2025) incorporated confidence-aware pose guidance to ensure smoother motion transitions and improve robustness against complex poses. Concurrently, the field has evolved toward Diffusion Transformer (DiT) architectures (Peebles & Xie, 2023), enabling the training of large-scale video generative models. Notable works include VACE (Jiang et al., 2025) and Wan-Animate (Cheng et al., 2025), which are based on the Wan (Wan et al., 2025) video foundational generative model.

Despite these remarkable advancements in global realism and temporal consistency, generating high-fidelity hand motions remains a persistent and unresolved challenge due to the highest motion amplitude and complexity of the hands. This stems from: **i)** the hands having the highest degrees of freedom compared to the head, torso, and legs, allowing for the largest range of motion; and **ii)** the presence of ten flexible fingers, which maximizes motion complexity (e.g., complex actions can rely solely on hands while other regions stay still). Therefore, generated videos often suffer from artifacts such as blur and malformations in the hands.

To mitigate this issue, Reinforcement Learning from Human Feedback (Christiano et al., 2017), and specifically Direct Preference Optimization (DPO) (Rafailov et al., 2023), provides a promising solution for aligning generative outputs with human preferences. Typically, DPO requires a dataset of preference pairs, i.e., distinct *winner (good)* and *loser (bad)* samples, to guide the optimization trajectory. The overall workflow for enhancing hand generation quality via the DPO paradigm typically involves the following steps. First, the pretrained model is used to generate several videos by different seeds under the same reference image and pose sequence. The generated videos are then manually annotated to select samples with high-quality hand generation (good samples) and those with low-quality hand genera-

[1]School of Artificial Intelligence, Beijing Normal University [2]Tencent Hunyuan. [*] Work done during the internship at Tencent Hunyuan. Correspondence to: Qinglin Lu <qinglinlu@tencent.com>, Zhen Cui <zhen.cui@bnu.edu.cn>.

*Proceedings of the 43rd International Conference on Machine Learning*, Seoul, South Korea. PMLR 306, 2026. Copyright 2026 by the author(s).

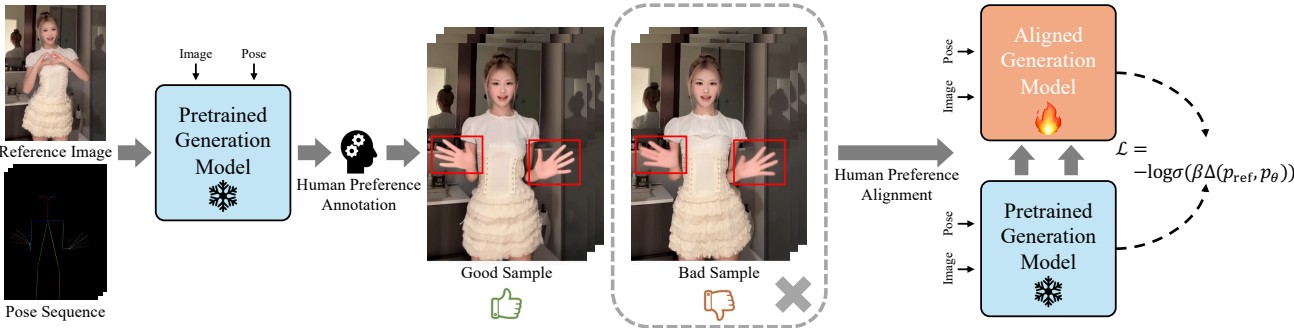

*Figure 1.* Overview of the Implicit Preference Alignment (IPA) framework for enhancing hand generation quality. IPA eliminates the necessity for bad samples inherent in standard preference optimization frameworks (e.g., direct preference optimization), alleviating the burden for preference annotation. We have also theoretically proved in Sec. 4.2 that IPA inherently performs implicit reward maximization.

tion (bad samples), forming good-bad preference pairs. As shown in Fig. 1, the good sample exhibits clear hand structure, whereas the bad sample suffers from blurring and distortion. Finally, these human preference pairs are utilized to conduct post-training for human preference alignment. While effective for static images or global video quality, applying DPO to improve dynamic hand generation presents a unique dilemma, i.e., constructing strict preference pairs for hands is prohibitively expensive and often impractical. This motivates our core inquiry: *Is it possible to lower the barrier for data construction and annotation while still maintaining effective preference alignment for hand regions?*

In this work, we challenge the necessity of strict preference pairs and propose *Implicit Preference Alignment (IPA)*, a novel and data-efficient post-training framework designed to enhance hand fidelity, as shown in Fig. 1. Our core observation is that although constructing rigorous preference pairs is difficult, obtaining isolated good samples remains relatively accessible and cost-effective. Theoretically grounded in implicit reward maximization, IPA eliminates the need for bad samples, which aligns the model by maximizing the likelihood of good samples while imposing a constraint to prevent deviation from the pretrained model. This formulation ensures that the model generalizes high-fidelity patterns from a limited set of good samples without suffering from mode collapse. In particular, we design a Hand-Aware Local Optimization mechanism to explicitly steer IPA toward hand regions, ensuring that the preference alignment process prioritizes these fine-grained structural details. Our main contributions are summarized as follows:

- We propose Implicit Preference Alignment, a data-efficient post-training framework that eliminates the need for strict preference pairs by aligning the model solely using self-generated high-quality samples.

- We introduce a Hand-Aware Local Optimization mechanism to explicitly steer the optimization process toward hand regions, effectively mitigating geometric distortions and blurring artifacts in complex motions.

- Extensive quantitative and qualitative experiments demonstrate that our method significantly enhances hand generation fidelity and overall video quality, outperforming existing state-of-the-art methods.

## 2. Related Work

The primary objective of human image animation is to synthesize high-fidelity, lifelike videos by driving a static reference image with a target pose sequence. This field has witnessed a significant paradigm shift with the evolution of generative networks. Initial approaches (Li et al., 2019; Siarohin et al., 2019; 2021; Zhao & Zhang, 2022) predominantly relied on Generative Adversarial Networks (GANs). These methods typically employ motion networks to estimate dense appearance flows, utilizing feature warping techniques to map the source appearance onto target poses. Despite their great success, GAN-based frameworks often struggle with training instability and mode collapse (Hu, 2024). Consequently, they frequently fail to maintain precise control over complex motions, resulting in synthesized videos plagued by visual artifacts.

Driven by the superior training stability and high-fidelity generation capabilities of the continuous-time modeling, recent research has largely pivoted toward diffusion models (Karras et al., 2023; Hu, 2024; Ma et al., 2024; Wang et al., 2024; Xu et al., 2024; Chang et al., 2024; Wang et al., 2025a; Zhang et al., 2025). Animate Anyone (Hu, 2024) designed a RefenceNet to extract human appearance features from the input image and align them with the motion generation branch. UniAnimate (Wang et al., 2025a) aligned reference image and video features within a shared space, employing a temporal Mamba (Gu & Dao, 2024) to achieve efficient human image animation. MimicMotion (Zhang et al., 2025) introduced confidence-aware pose guidance to ensure high frame quality and proposed hand region enhancement to alleviate hand distortion.

More recently, the emergence of DiT-based large model architectures (Kong et al., 2024; Yang et al., 2025; Wan

et al., 2025) has significantly advanced video generation capabilities. The adaptation of these models for human image animation has yielded marked improvements in both character realism and temporal consistency. For example, UniAnimate-DiT (Wang et al., 2025b) extended Uni-Animate to the Wan2.1 (Wan et al., 2025) video foundational generative model. As an all-in-one video generation model, VACE (Jiang et al., 2025) was built upon the Wan2.1 and underwent extensive training and expansion using vast amounts of data, enabling seamless support for human image animation. Wan-Animate (Cheng et al., 2025) proposed a unified framework for image animation and replacement.

## 3. Preliminaries

### 3.1. Generative Modeling via Flow Matching

Flow matching aims to transform a source distribution $p_0$ to a target distribution $p_1$ via a continuous-time vector field (Lipman et al., 2023; Liu et al., 2023). In the context of Rectified Flow (Liu et al., 2023), the probability path is defined as a linear interpolation between the source and target. Let $\mathbf{Z}_0 \sim p_0$ and $\mathbf{Z}_1 \sim p_1$, the intermediate state $\mathbf{Z}_t$ at timestep $t \in [0, 1]$ is defined as:

$$\mathbf{Z}_t = t\mathbf{Z}_1 + (1 - t)\mathbf{Z}_0. \tag{1}$$

This path corresponds to a constant velocity field $v(\mathbf{Z}_t, t) = \mathbf{Z}_1 - \mathbf{Z}_0$. The generative model $v_\theta$ is trained to approximate this velocity field by minimizing the mean squared error:

$$\mathcal{L}_{\text{FM}} = \mathbb{E}_{t\sim\mathcal{U}(0,1),\mathbf{z}_0,\mathbf{z}_1}[\|v_\theta(\mathbf{Z}_t; t, c) - (\mathbf{Z}_1 - \mathbf{Z}_0)\|_2^2], \tag{2}$$

where $c$ represents the conditional information (e.g., text prompt, reference image). Benefiting from its training stability and efficient straight-line inference paths, Flow Matching has emerged as a fundamental generative paradigm widely adopted for image and video generation tasks (Esser et al., 2024; Kong et al., 2024; Labs et al., 2025; Wan et al., 2025).

### 3.2. Reinforcement Learning from Human Feedback

Reinforcement Learning from Human Feedback (RLHF) aligns models with human preferences by maximizing a reward signal while restraining the model from deviating largely from the initial pretrained model (Christiano et al., 2017; Kupcsik et al., 2017; Ziegler et al., 2019). Let $\pi_{\text{ref}}$ denote the reference policy and $\pi_\theta$ the policy to be optimized. Based on (Jaques et al., 2017; 2020), the standard RLHF objective is formulated as:

$$\max_{\pi_\theta} \mathbb{E}_{x,y}\left[r(x, y)\right] - \beta D_{\text{KL}}(\pi_\theta(y|x)\|\pi_{\text{ref}}(y|x)), \tag{3}$$

where $r(x, y)$ is the reward function derived from human preferences, and $\beta$ is a coefficient controlling the strength of the KL-divergence penalty. Direct Preference Optimization

(DPO) (Rafailov et al., 2023) further simplifies this by directly optimizing the policy using preference pairs $(y_w, y_l)$, bypassing the explicit reward modeling step. Benefiting from its simplicity, DPO has been widely applied in the field of image and video generation, evolving into variants based on different generative paradigms such as Diffusion-DPO (Wallace et al., 2024), Flow-DPO (Liu et al., 2025).

## 4. Method

### 4.1. Problem Formulation

**Problem.** Let $I$ and $\mathcal{P}$ denote a static human image and a sequence of poses, respectively. The goal of human image animation is to generate a dynamic video $\mathcal{V}$ with continuous motion under the condition of $I$ and $\mathcal{P}$. The generation process can be formalized as:

$$\mathcal{V} = \mathcal{G}\left(\mathbf{Z} \sim \mathcal{N}(\mu, \sigma^2), I, \mathcal{P}\right), \tag{4}$$

where $\mathcal{G}$ denotes a large-scale dynamic video generator (e.g., VACE (Jiang et al., 2025)), and $\mathbf{Z}$ represents a prior state sampled from the Gaussian prior distribution.

Compared to general video generation tasks, human image animation typically exhibits higher motion dynamics. This is because the character in the reference image is required to perform diverse actions conditioned on pose signals. Especially for the hand region, due to its high degree of freedom and complexity in movement, generated videos often exhibit distortion and collapse of the hands. Therefore, enhancing the fidelity of hand has emerged as a critical focal point in this field (Zhang et al., 2025).

To enhance the fidelity of hand regions, Reinforcement Learning from Human Feedback (RLHF) offers a promising avenue for preference alignment. Direct Preference Optimization (DPO) (Rafailov et al., 2023) is an efficient choice that bypasses an explicit reward model by performing direct alignment using self-generated preference pairs (i.e., good-bad samples) annotated by humans. While DPO offers an efficient simplification of RLHF, it faces substantial challenges when targeting hand region quality. The construction of preference pairs is considerably more intricate and costly than in general video tasks, largely due to the frame-wise inconsistency of hand states. To illustrate this, we outline four potential scenarios for defining preference pairs between two generated videos, $\mathcal{V}_A$ and $\mathcal{V}_B$:

**Case 1:** Both $\mathcal{V}_A$ and $\mathcal{V}_B$ consistently satisfy human preference standards across every frame.

**Case 2:** Both $\mathcal{V}_A$ and $\mathcal{V}_B$ consistently fail to meet human preference standards in any frame.

**Case 3:** Both videos exhibit mixed quality, where some frames satisfy human preference while others do not.

**Case 4:** $\mathcal{V}_A$ consistently satisfies human preference standards in every frame, whereas $\mathcal{V}_B$ fails.

Crucially, Case 4 is the only scenario compliant with DPO. In other words, even if *good* samples are successfully sampled, the inability to consistently sample valid *bad* counterparts renders the application of DPO impractical.

**Main Idea.** The core idea of this work is to design a preference optimization framework that relies solely on *good* samples (i.e., **Case 1**). This strategy directly reduces data production costs by obviating the need to curate strict preference pairs with distinct quality differences. To achieve this, our approach must satisfy two critical prerequisites: **i)** the model needs to extract and generalize high-fidelity generation patterns from self-generated good samples; and **ii)** we must avoid mode collapse to ensure the model does not forget the large-scale pre-trained knowledge acquired during its initial training. We refer to this framework as *Implicit Preference Alignment*.

## 4.2. Implicit Preference Alignment

We define $p_{\text{ref}}$ as the pretrained reference model that encapsulates vast general knowledge, and $p_\theta$ as the preference-aligned model to be optimized for generalizing high-fidelity patterns from a limited set of good samples. We denote the data distribution of preference samples as $q(\mathbf{X})$.

**Objective 1:** We expect $p_\theta$ to match the preferred data distribution $q(\mathbf{X})$ better than $p_{\text{ref}}$. Thus, we have:

$$D_{\text{KL}}(q(\mathbf{X})||p_\theta(\mathbf{X})) < D_{\text{KL}}(q(\mathbf{X})||p_{\text{ref}}(\mathbf{X})). \quad (5)$$

This inequality implies that the distributional discrepancy between $p_\theta$ and $q(\mathbf{X})$ must be strictly smaller than that between $p_{\text{ref}}$ and $q(\mathbf{X})$. Since the preceding distributions are intractable, we follow (Wallace et al., 2024) and leverage the continuous-time latent trajectory $\mathbf{Z}_{0:1}$ for approximation:

$$\begin{aligned} D_{\text{KL}}(q(\mathbf{Z}_{0:1}|\mathbf{X})||p_\theta(\mathbf{Z}_{0:1}|I,\mathcal{P})) \\ < D_{\text{KL}}(q(\mathbf{Z}_{0:1}|\mathbf{X})||p_{\text{ref}}(\mathbf{Z}_{0:1}|I,\mathcal{P})). \end{aligned} \quad (6)$$

For notational simplicity, we abbreviate $q(\mathbf{Z}_{0:1}|\mathbf{X})$, $p_\theta(\mathbf{Z}_{0:1}|I,\mathcal{P})$, and $p_{\text{ref}}(\mathbf{Z}_{0:1}|I,\mathcal{P})$ as $q$, $p_\theta$, and $p_{\text{ref}}$, respectively. Rearranging the terms of the above inequality yields:

$$D_{\text{KL}}(q||p_{\text{ref}}) - D_{\text{KL}}(q||p_\theta) > 0. \quad (7)$$

We further define the above KL divergence gap as:

$$\Delta(p_{\text{ref}}, p_\theta) = D_{\text{KL}}(q||p_{\text{ref}}) - D_{\text{KL}}(q||p_\theta). \quad (8)$$

Substituting this into Eq. (7) yields:

$$\Delta(p_{\text{ref}}, p_\theta) > 0. \quad (9)$$

This implies that to fulfill **Objective 1**, we must ensure the KL divergence gap is positive. To enforce this positivity, we

formulate the following log-sigmoid loss function:

$$\mathcal{L} = -\log\sigma(\Delta(p_{\text{ref}}, p_\theta)). \quad (10)$$

Intuitively, this objective function employs a penalty mechanism that compels the model to learn parameters satisfying $\Delta(p_{\text{ref}}, p_\theta) > 0$. Specifically, when $\Delta(p_{\text{ref}}, p_\theta) < 0$, the loss incurs a sharp increase. Thus, the optimization process drives the model to adjust its parameters to minimize the loss, ultimately stabilizing $\Delta(p_{\text{ref}}, p_\theta)$ at a positive value.

While the aforementioned objective ensures that $p_\theta$ outperforms $p_{\text{ref}}$ by closely approximating the preference distribution $q(\mathbf{X})$, optimization should not be excessive. We must avoid over-fitting to the limited preference data, which risks causing catastrophic forgetting of the pretrained knowledge.

**Objective 2:** Ensuring preference alignment without over-fitting, we impose a constraint coefficient $\beta$ on Eq. (10):

$$\mathcal{L} = -\log\sigma(\beta\Delta(p_{\text{ref}}, p_\theta)). \quad (11)$$

The core of $\beta$ is to quantify the permissible deviation of the preference-aligned model $p_\theta$ from the reference model $p_{\text{ref}}$. By modulating the penalty strength on this divergence, it indirectly controls overfitting during fine-tuning. Specifically, a larger $\beta$ imposes a stricter constraint on the deviation, keeping $p_\theta$ closer to $p_{\text{ref}}$; conversely, a smaller $\beta$ relaxes the constraint, allowing for larger deviation. Moreover, an equally valid and insightful interpretation emerges when examining the training dynamics through the log-sigmoid function. In this view, $\beta$ dictates the steepness of the sigmoid curve, effectively controlling the gradient saturation speed. The underlying mechanism is likely a synergistic combination of both effects, which remains an open issue not definitively resolved in this work.

**Theoretical Insights:** *Fundamentally, Eq. (11) serves as a surrogate to optimize an implicit reward function. It navigates the trade-off between maximizing the alignment of generated videos with preference data and minimizing the divergence from the pretrained model. That is, the goal is to maximize consistency with human preferences without deviating excessively from the pretrained priors.* Next, we provide a theoretical justification for this claim.

**Theoretical Analysis:** Let $r(\mathbf{X}, I, \mathcal{P}; \mathcal{D}_{\text{pref}}) := r(\mathbf{X}, I, \mathcal{P})$ denote a reward function designed to quantify the preference consistency between the generated sample $\mathbf{X}$ and the preference dataset $\mathcal{D}_{\text{pref}}$, conditioned on the reference image $I$ and the pose sequence $\mathcal{P}$. Our objective is to identify the optimal policy $p_\theta$ that achieves high preference consistency for generated videos, while simultaneously maintaining minimal deviation from $p_{\text{ref}}$. Based on Eq. (3), the RLHF objective in this scenario is formulated as:

$$\begin{aligned} \max_{p_\theta} \mathbb{E}_{\mathbf{X}, I, \mathcal{P}}[r(\mathbf{X}, I, \mathcal{P})] \\ - \beta D_{\text{KL}}(p_\theta(\mathbf{X}|I,\mathcal{P})||p_{\text{ref}}(\mathbf{X}|I,\mathcal{P})). \end{aligned} \quad (12)$$

Following (Wallace et al., 2024), we further approximate this objective via $\mathbf{Z}_{0:1}$ as:

$$\max_{p_\theta} \mathbb{E}_{\mathbf{Z}_{0:1}, I, \mathcal{P}}[r(\mathbf{Z}_{0:1}, I, \mathcal{P})] \tag{13}$$
$$- \beta D_{\mathrm{KL}}(p_\theta(\mathbf{Z}_{0:1}|I, \mathcal{P})||p_{\mathrm{ref}}(\mathbf{Z}_{0:1}|I, \mathcal{P})).$$

Following prior works (Peters & Schaal, 2007; Peng et al., 2019; Korbak et al., 2022; Go et al., 2023; Rafailov et al., 2023), the optimal solution to the KL-constrained reward maximization objective in Eq. (13) takes the following form:

$$p_\theta(\mathbf{Z}_{0:1}|I, \mathcal{P}) = \frac{1}{Z} p_{\mathrm{ref}}(\mathbf{Z}_{0:1}|I, \mathcal{P}) \exp\left(\frac{r(\mathbf{Z}_{0:1}, I, \mathcal{P})}{\beta}\right), \tag{14}$$

where $Z$ is a normalization constant that does not depend on $\mathbf{Z}_{0:1}$. For notational brevity, we rewrite this as:

$$p_\theta = \frac{1}{Z} p_{\mathrm{ref}} \exp\left(\frac{r}{\beta}\right). \tag{15}$$

Taking the logarithm of both sides of the equation yields:

$$\log p_\theta = \log p_{\mathrm{ref}} + \frac{r}{\beta} - \log Z. \tag{16}$$

Rearranging the equation yields the reward function $r$:

$$r = \beta(\log p_\theta - \log p_{\mathrm{ref}}) + \beta \log Z. \tag{17}$$

Focusing on the expected performance over the preference distribution $q$, we take the expectation $\mathbb{E}_q$ on both sides:

$$\mathbb{E}_q[r] = \beta\mathbb{E}_q[\log p_\theta - \log p_{\mathrm{ref}}] + \beta\mathbb{E}_q[\log Z]. \tag{18}$$

According to the definition of KL divergence, i.e.,

$$D_{\mathrm{KL}}(q||p_\theta) = \mathbb{E}_q[\log q] - \mathbb{E}_q[\log p_\theta], \tag{19}$$

$$D_{\mathrm{KL}}(q||p_{\mathrm{ref}}) = \mathbb{E}_q[\log q] - \mathbb{E}_q[\log p_{\mathrm{ref}}]. \tag{20}$$

We have:

$$\begin{aligned}
\mathbb{E}_q[r] &= \beta\mathbb{E}_q[\log p_\theta - \log p_{\mathrm{ref}}] + \beta\mathbb{E}_q[\log Z] \\
&= \beta(\mathbb{E}_q[\log p_\theta] - \mathbb{E}_q[\log p_{\mathrm{ref}}]) + \beta\mathbb{E}_q[\log Z] \\
&= \beta(D_{\mathrm{KL}}(q||p_{\mathrm{ref}}) - D_{\mathrm{KL}}(q||p_\theta)) + \beta\mathbb{E}_q[\log Z] \\
&= \beta\Delta(p_{\mathrm{ref}}, p_\theta) + \beta\mathbb{E}_q[\log Z].
\end{aligned} \tag{21}$$

By defining the constant $\beta\mathbb{E}_q[\log Z] = C$, we obtain the complete formulation:

$$\mathbb{E}_{q(\mathbf{Z}_{0:1}|\mathbf{X})}[r(\mathbf{Z}_{0:1}, I, \mathcal{P})] = \beta\Delta(p_{\mathrm{ref}}, p_\theta) + C. \tag{22}$$

This equation establishes that maximizing $\beta\Delta(p_{\mathrm{ref}}, p_\theta)$ is equivalent to maximizing the reward. Furthermore, it shows that minimizing $\mathcal{L} = -\log\sigma(\beta\Delta(p_{\mathrm{ref}}, p_\theta))$ is also equivalent to reward maximization. *Consequently, we have provided theoretical justification that our objective function inherently optimizes an implicit reward function.*

### 4.3. Flow IPA

In practice, directly computing $\Delta(p_{\mathrm{ref}}, p_\theta)$ is computationally intractable, as it necessitates evaluating the likelihood across all continuous timesteps. Consequently, we must reformulate it into a tractable form. Leveraging insights from (Kingma & Gao, 2023; Liu et al., 2025), the KL divergence term of $\Delta(p_{\mathrm{ref}}, p_\theta)$ within the flow matching paradigm (Liu et al., 2023) can be formalized as:

$$\frac{\mathrm{d}}{\mathrm{d}t} D_{\mathrm{KL}}(q(\mathbf{Z}_{t:1}|\mathbf{X})||p_{\mathrm{ref}}(\mathbf{Z}_{t:1}|I, \mathcal{P})) \tag{23}$$
$$= \frac{1}{2}(1-t)^2 \mathbb{E}_v[\|v - v_{\mathrm{ref}}(\mathbf{Z}_t; t, I, \mathcal{P})\|_2^2],$$

$$\frac{\mathrm{d}}{\mathrm{d}t} D_{\mathrm{KL}}(q(\mathbf{Z}_{t:1}|\mathbf{X})||p_\theta(\mathbf{Z}_{t:1}|I, \mathcal{P})) \tag{24}$$
$$= \frac{1}{2}(1-t)^2 \mathbb{E}_v[\|v - v_\theta(\mathbf{Z}_t; t, I, \mathcal{P})\|_2^2].$$

where $v = \mathbf{Z}_1 - \mathbf{Z}_0$. $v_\theta$ and $v_{\mathrm{ref}}$ are two continuous-time velocity field models. Therefore, we have:

$$\frac{\mathrm{d}}{\mathrm{d}t}\Delta_t(p_{\mathrm{ref}}, p_\theta) = \frac{1}{2}(1-t)^2 \mathbb{E}_v[\|v - v_{\mathrm{ref}}(\mathbf{Z}_t; t, I, \mathcal{P})\|_2^2 \tag{25}$$
$$- \|v - v_\theta(\mathbf{Z}_t; t, I, \mathcal{P})\|_2^2],$$

We derive the total deviation $\Delta(p_{\mathrm{ref}}, p_\theta)$ by integrating across the time interval $t \in [0, 1]$:

$$\begin{aligned}
\Delta(p_{\mathrm{ref}}, p_\theta) &= \int_0^1 \frac{\mathrm{d}}{\mathrm{d}t}\Delta_t(p_{\mathrm{ref}}, p_\theta)\mathrm{d}t \\
&= \int_0^1 \frac{1}{2}(1-t)^2 \mathbb{E}_v[\|v - v_{\mathrm{ref}}(\mathbf{Z}_t; t, I, \mathcal{P})\|_2^2 \\
&\quad - \|v - v_\theta(\mathbf{Z}_t; t, I, \mathcal{P})\|_2^2]\mathrm{d}t \\
&= \mathbb{E}_{t\sim\mathcal{U}(0,1),v}[\frac{1}{2}(1-t)^2(\|v - v_{\mathrm{ref}}(\mathbf{Z}_t; t, I, \mathcal{P})\|_2^2 \\
&\quad - \|v - v_\theta(\mathbf{Z}_t; t, I, \mathcal{P})\|_2^2)].
\end{aligned} \tag{26}$$

Substituting the above equation into Eq. (11) yields:

$$\begin{aligned}
\mathcal{L} = \ \mathbb{E}_{t\sim\mathcal{U}(0,1),v}[&-\log\sigma(\frac{\beta}{2}(1-t)^2(\|v - v_{\mathrm{ref}}(\mathbf{Z}_t; t, I, \mathcal{P})\|_2^2 \\
&- \|v - v_\theta(\mathbf{Z}_t; t, I, \mathcal{P})\|_2^2))].
\end{aligned} \tag{27}$$

### 4.4. Hand-Aware Local Optimization

To explicitly steer the preference alignment towards hand regions, we propose a hand-aware local optimization mechanism. We first construct a spatial weight matrix $\mathbf{W}$:

$$\mathbf{W} = \mathbf{1} + \lambda \cdot \mathbf{M}, \tag{28}$$

where $\mathbf{M}$ denotes the binary mask of the hand regions, and $\lambda$ represents the hand enhancement coefficient. Note that

the binary hand mask $\mathbf{M}$ can be directly derived from the hand keypoint coordinates within the pose sequence.

By injecting $\mathbf{W}$ into Eq. (27), we obtain the final weighted optimization objective:

$$
\begin{aligned}
\mathcal{L} = \mathbb{E}_{t \sim \mathcal{U}(0,1), v} \big[ \\
- \log \sigma(\frac{\beta}{2}(1-t)^2 (\| \sqrt{\mathbf{W}} \odot (v - v_{\mathrm{ref}}(\mathbf{Z}_t; t, I, \mathcal{P})) \|_2^2 \\
- \| \sqrt{\mathbf{W}} \odot (v - v_\theta(\mathbf{Z}_t; t, I, \mathcal{P})) \|_2^2)) \big].
\end{aligned}
\tag{29}
$$

This weighted objective empowers the implicit preference alignment to prioritize the improvement of hand quality.

# 5. Experiments

## 5.1. Implementation Details

Our framework utilizes the DiT-based generative model VACE-14B (Jiang et al., 2025) as our pretrained model, which is an all-in-one video generation model endowed with large-scale prior knowledge. To curate preference data, we first collect 1,500 human dancing videos from the Internet. We then use DWPose (Yang et al., 2023) to extract pose sequences from each video and randomly sample one frame as the reference image. Finally, we employ VACE to generate 6,000 candidate videos (four samples per pose-image pair), from which 93 high-quality samples are meticulously hand-picked through a stringent human filtering process for subsequent training. All generated videos have a spatial resolution of $832 \times 480$ and a temporal length of 81 frames. Following prior work (Liu et al., 2025), we use the LoRA (Hu et al., 2022) training mode with rank 128 (applied only to the QKV projections) to fit these preference data. The whole framework is trained on 8 NVIDIA H20 GPUs with a batch size of 8. Based on empirical results, the hyperparameters $\beta$ and $\lambda$ are set to 600 and 10, respectively. The entire optimization process spans 1,000 training steps.

**Evaluation details.** Following previous work (Zhang et al., 2025), we adopt the TikTok (Jafarian & Park, 2021) dataset and use sequence 335 to 340 for our evaluation. To further facilitate a more comprehensive evaluation, we construct a more challenging benchmark. Specifically, this benchmark comprises 100 curated cases covering a wide spectrum of complex hand dynamics (e.g., intricate finger dance). Crucially, these samples are strictly disjoint from the training set to ensure a fair evaluation of the model's generalization capability. We consider four standard evaluation metrics that are used in (Zhang et al., 2025), including: FID-VID (Balaji et al., 2019), FVD (Unterthiner et al., 2019), SSIM (Wang et al., 2004), and Peak Signal-to-Noise Ratio (PSNR).

## 5.2. Baseline Comparisons

We compare our method with the current state-of-the-art methods, including four image generative model-based methods (MagicAnimate (Xu et al., 2024), MagicPose (Chang et al., 2024), Moore-AnimateAnyone (MooreThreads, 2024), MuseV (Xia et al., 2024)) and five video generative model-based methods (MimicMotion (Zhang et al., 2025), UniAnimate-DiT (Wang et al., 2025b), VACE (Jiang et al., 2025), Wan2.2-Fun-A14B-Control (Alibaba-PAI, 2025), Wan-Animate (Cheng et al., 2025)). Specifically, MagicAnimate, MagicPose, Moore-AnimateAnyone, and MuseV are built upon Stable Diffusion v1.5 (Rombach et al., 2022); MimicMotion is based on Stable Video Diffusion (Blattmann et al., 2023); while UniAnimate-DiT, VACE, Wan2.2-Fun-A14B-Control, and Wan-Animate are derived from the Wan (Wan et al., 2025) foundational model. Notably, for our benchmark, we only compare our framework against recent video generative model-based methods. We exclude image-based models from this specific evaluation, as their inherent architectural limitations in maintaining temporal consistency make it inequitable to assess them on scenarios involving highly complex hand dynamics.

Table 1. Quantitative comparison on the TikTok benchmark.

| Methods | FID-VID ↓ | FVD ↓ | SSIM ↑ | PSNR ↑ |
|---|---|---|---|---|
| MagicAnimate | 16.2 | 848 | 0.740 | 17.5 |
| MagicPose | 13.3 | 916 | 0.776 | 18.8 |
| Moore-AnimateAnyone | 12.4 | 728 | 0.758 | 18.7 |
| MuseV | 14.6 | 754 | 0.766 | 17.6 |
| MimicMotion | 9.3 | 594 | 0.795 | 20.1 |
| UniAnimate-DiT | 11.8 | 350 | 0.768 | 20.4 |
| VACE | 13.4 | 427 | 0.777 | 20.2 |
| Wan2.2-Fun-A14B-Control | 9.8 | 360 | 0.773 | 20.7 |
| Wan-Animate | 8.6 | 316 | 0.799 | 20.5 |
| Ours | **5.9** | **255** | **0.841** | **23.8** |

Table 2. Quantitative comparison on our benchmark.

| Methods | FID-VID ↓ | FVD ↓ | SSIM ↑ | PSNR ↑ |
|---|---|---|---|---|
| MimicMotion | 14.6 | 420 | 0.577 | 15.7 |
| UniAnimate-DiT | 10.6 | 293 | 0.646 | 17.4 |
| VACE | 12.5 | 327 | 0.668 | 18.2 |
| Wan2.2-Fun-A14B-Control | 11.7 | 296 | 0.658 | 17.4 |
| Wan-Animate | 13.6 | 376 | 0.703 | 17.3 |
| Ours | **6.3** | **224** | **0.757** | **21.5** |

Table 3. Quantitative comparison on hand regions.

| Methods | SSIM-Hand ↑ | PSNR-Hand ↑ |
|---|---|---|
| MimicMotion | 0.444 | 12.7 |
| UniAnimate-DiT | 0.472 | 14.2 |
| VACE | 0.501 | 15.3 |
| Wan2.2-Fun-A14B-Control | 0.507 | 14.3 |
| Wan-Animate | 0.544 | 14.1 |
| Ours | **0.606** | **18.9** |

**Quantitative results.** As shown in Tab. 1, our method achieves the best performance across all evaluation metrics on the TikTok benchmark. Specifically, compared to the strongest competitor Wan-Animate, our method significantly reduces FID-VID from 8.6 to 5.9 and FVD from 316

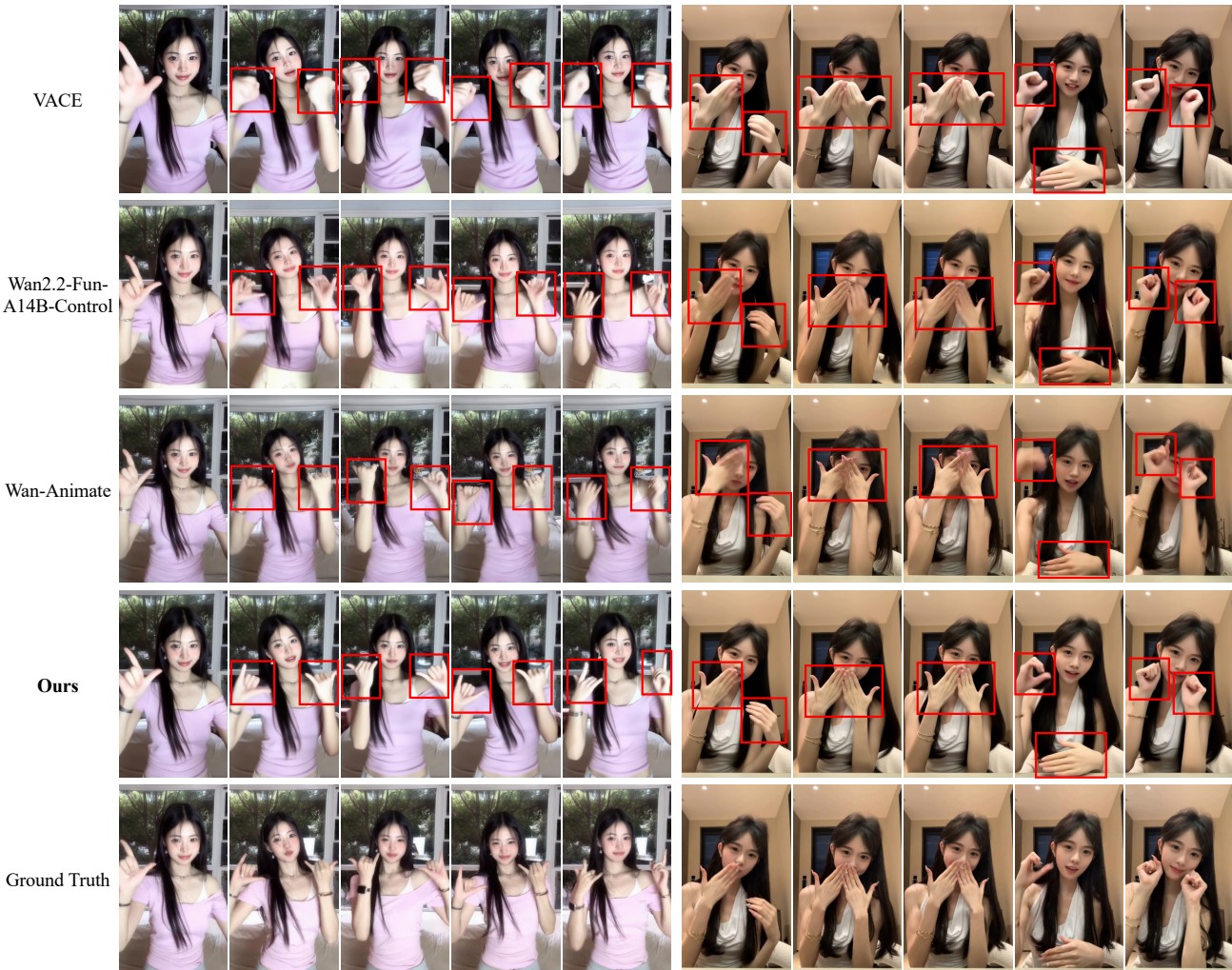

*Figure 2.* Visual comparisons of different methods. Existing methods often suffer from malformed or collapsed hand appearances. In contrast, our approach yields hands with sharp edges and distinct finger separation, closely matching the Ground Truth. Complete comparisons can be found in Fig. 7.

to 255. Furthermore, our method achieves the highest scores in structural metrics, with the SSIM of 0.841 and PSNR of 23.8, indicating a substantial improvement in frame-wise fidelity. The advantages of our framework are even more pronounced on our proposed challenging benchmark, which focuses on complex hand dynamics, as shown in Tab. 2. This empirical evidence confirms that our IPA, combined with Hand-Aware Local Optimization, effectively steers the model to generate high-fidelity details even in scenarios with complex hand dynamics, outperforming existing baselines.

**Quantitative results on hands.** Regarding the quantitative evaluation of hand regions, since the field prioritizes overall generation quality, there are no standard metrics specifically for hand regions. Consequently, recent approaches like MimicMotion depend entirely on qualitative visual analysis to evaluate their hand region enhancement. To quantitatively evaluate hand generation quality, we leverage hand masks to measure two pixel-wise and frame-wise metrics,

termed SSIM-Hand and PSNR-Hand. Tab. 3 lists the quantitative comparison of different methods on hand regions. We can observe that our method consistently outperforms all baseline models on both metrics. These results quantitatively demonstrate the effectiveness of our framework in preserving hand structural integrity and texture details.

**Qualitative results.** Fig. 2 provides some visual comparisons between our method and state-of-the-art baselines. We can first observe that generating high-fidelity hand motions remains a significant challenge for existing methods, frequently exhibiting blurred structures and geometric distortions in hand regions. For example, during complex finger dance sequences where hand dynamics change rapidly, these models often fail to maintain structural integrity, leading to malformed or collapsed hand appearances. In contrast, our method significantly improves the perceptual quality of hand generation. By leveraging IPA to learn from high-quality samples, our model successfully generates clear and

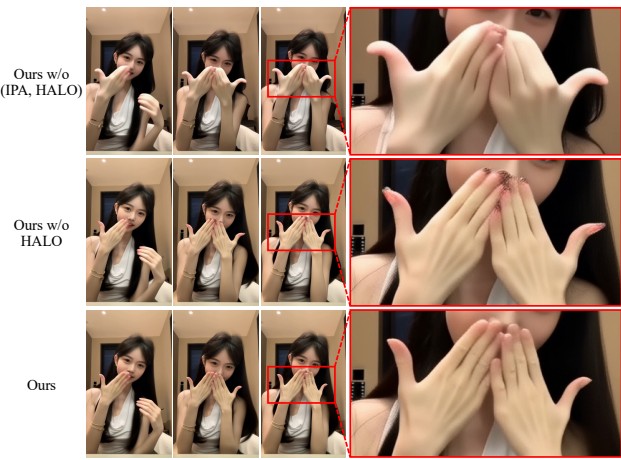

*Figure 3.* Visual results of ablation study for key components.

anatomically correct hand structures even under challenging motion conditions.

### 5.3. Ablation Studies

We evaluate the effects of the key components in our method, including IPA and Hand-Aware Local Optimization (HALO). The results are presented in Tab. 4, from which we draw the following conclusions: **i)** IPA is effective and yields substantial performance improvements, as it can align the model with human preferences by using self-generated high-quality samples. **ii)** The inclusion of HALO yields further improvements, confirming the feasibility and effectiveness of explicitly steering the optimization toward hand regions. Furthermore, we provide the visual results of ablation studies for key components in Fig. 3. We can observe that *Ours w/o (IPA, HALO)* exhibits severe malformations and distortions in hand regions. *Ours w/o HALO* alleviates geometric distortions; however, it still suffers from blurry artifacts. In contrast, *Ours* produces superior results with distinct hand structures and texture details. These visual results further demonstrate the effectiveness of IPA and HALO in improving hand generation quality.

*Table 4.* Ablation study of the key components in our method.

| Dataset | IPA | HALO | FID-VID ↓ | FVD ↓ | SSIM ↑ | PSNR ↑ |
|---------|-----|------|-----------|-------|--------|--------|
| | ✓ | ✓ | **5.9** | **255** | **0.841** | **23.8** |
| TikTok | ✓ | ✗ | 7.9 | 288 | 0.819 | 22.7 |
| | ✗ | ✗ | 13.4 | 427 | 0.777 | 20.2 |
| | ✓ | ✓ | **6.3** | **224** | **0.757** | **21.5** |
| Ours | ✓ | ✗ | 8.1 | 251 | 0.741 | 20.6 |
| | ✗ | ✗ | 12.5 | 327 | 0.668 | 18.2 |

**Exploring the effects of different $\beta$.** We conduct the ablation studies to explore the effects of different $\beta$. Fig. 4 illustrates the performance trends across varying $\beta$ values, we can observe the following phenomena: **i)** When $\beta$ is small (i.e., $\beta = 200$), the insufficient constraint makes the model

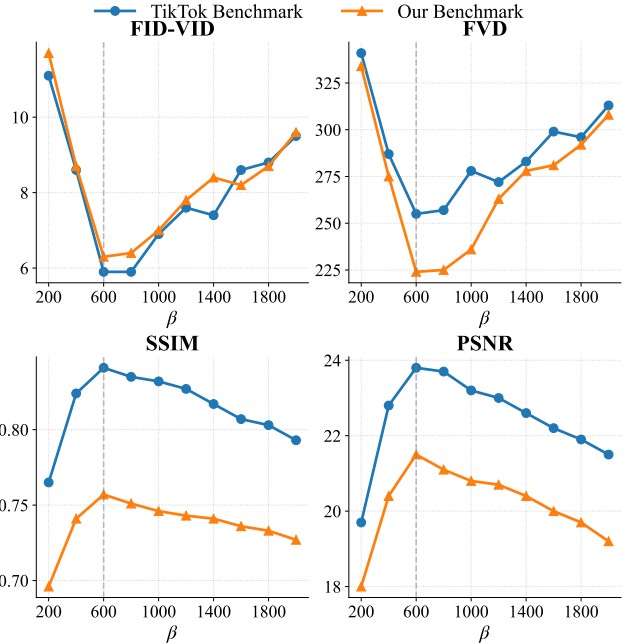

*Figure 4.* Ablation study on different $\beta$. We can observe that the optimal performance is achieved when $\beta = 600$.

*Table 5.* Ablation study on different $\lambda$.

| Dataset | $\lambda$ | FID-VID ↓ | FVD ↓ | SSIM ↑ | PSNR ↑ |
|---------|-----------|-----------|-------|--------|--------|
| | 0.1 | 7.4 | 283 | 0.827 | 22.9 |
| TikTok | 1.0 | 6.9 | 278 | 0.831 | 23.0 |
| | 10 | **5.9** | **255** | **0.841** | **23.8** |
| | 100 | 6.8 | 265 | 0.832 | 23.2 |
| | 0.1 | 7.8 | 239 | 0.743 | 20.7 |
| Ours | 1.0 | 7.5 | 234 | 0.745 | 20.9 |
| | 10 | **6.3** | **224** | **0.757** | **21.5** |
| | 100 | 6.9 | 225 | 0.755 | 21.3 |

prone to overfitting, thereby deteriorating performance. **ii)** As $\beta$ increases, performance gradually improves, peaking at $\beta = 600$. **iii)** Beyond $\beta = 600$, performance begins to decline as $\beta$ increases further. This is attributed to the overly strict constraint imposed by an excessive $\beta$, which hinders the model from effectively learning the high-fidelity patterns from the good samples.

**Exploring the effects of different $\lambda$.** We investigate the effects of the weighting coefficient $\lambda$ in the HALO mechanism, which controls the focus on hand regions. As shown in Tab. 5, increasing $\lambda$ from 0.1 to 10 leads to continuous improvements in all metrics. However, setting $\lambda$ too large (i.e., 100) results in a slight performance saturation or degradation. This suggests that while emphasizing hands is crucial, an excessive weight might disrupt the global quality of the video. Therefore, we adopt $\lambda = 10$ as the optimal setting.

## 6. Broader Discussion for IPA and DPO

In this section, we provide a broader discussion comparing our proposed IPA with the standard DPO.

## 6.1. Structural Comparison and Novelty Positioning

Let us directly compare the formulas. If we take the standard Flow-DPO objective and simply drop the negative sample term, the resulting expression is:

$$\mathcal{L}_{\text{Pos-DPO}} = \mathbb{E}[-\log \sigma(\beta(\|v_w - v_{\text{ref}}\|_2^2 - \|v_w - v_\theta\|_2^2))]. \quad (30)$$

Our IPA objective is:

$$\mathcal{L}_{\text{IPA}} = \mathbb{E}[-\log \sigma(\frac{\beta}{2}(1-t)^2(\|(v_w - v_{\text{ref}})\|_2^2 - \|(v_w - v_\theta)\|_2^2))]. \quad (31)$$

The structural differences and novelty positioning:

- **Structure is Equivalent, Derivation is Novel.** We observe that the structural form is mathematically equivalent. However, Flow-DPO borrows this structure from the Bradley-Terry model. In contrast, our contribution derives this exact form from first principles: minimizing the KL-divergence gap between the preference distribution and the model, under a strict prior constraint.

- **Our Novelty Positioning.** We do not claim novelty in inventing a new algebraic operator. Rather, our contribution lies in the theoretical and practical justification for why this reduction is not only viable but essential for complex generation tasks.

## 6.2. Data Constraints and Comparison Fairness

A primary motivation for IPA is the difficulty of constructing strict preference pairs for dynamic hand motions. To quantify this challenge, we analyze our curated dataset. Among the 93 high-quality samples used for our IPA, we attempted to identify corresponding bad samples to form valid DPO pairs, and only 7 samples (approximately 7.5%) could be paired. This scarcity creates a dilemma for a direct and fair comparison: **i)** Comparing IPA (trained on 93 samples) against DPO (trained on only 7 pairs) would be inequitable due to the vast disparity in training data volume. **ii)** Conversely, generating enough valid preference pairs to match the IPA dataset size would incur high computational and annotation costs, undermining the premise of data efficiency. Thus, we emphasize that a direct comparison under identical cost and sample size conditions is not feasible.

## 6.3. Positioning IPA and DPO

It is important to clarify that we do not claim IPA is inherently superior to DPO in all general scenarios. DPO benefits significantly from explicit negative signals provided by bad samples, which can effectively push the model away from undesirable behaviors. However, this relies heavily on the availability of high-quality paired data. Our work positions IPA as a specialized, resource-efficient alternative designed for scenarios where high-quality preference pairs are scarce.

## 6.4. A Strategic Trade-off

We suggest that the choice between IPA and DPO represents a trade-off based on task complexity and resource availability:

- **DPO is preferable when:** The task is relatively simple (making it easy to distinguish and generate good/bad pairs), or when resources allow for extensive data generation and annotation. In these cases, the explicit negative feedback from DPO can provide a robust optimization signal.

- **IPA is preferable when:** The task is highly complex (e.g., dynamic hand articulation with high degrees of freedom) or limited available resources. In such resource-constrained or data-scarce environments, IPA offers a highly efficient pathway to preference alignment by leveraging only self-generated good samples.

## 7. Conclusion

In this paper, we have proposed Implicit Preference Alignment (IPA), a novel and data-efficient post-training framework designed to address the persistent challenge of generating high-fidelity hand motions in human image animation. By theoretically deriving an implicit reward maximization objective, IPA eliminates the expensive requirement for constructing strict preference pairs, allowing the model to be aligned solely using good samples. Moreover, we have introduced Hand-Aware Local Optimization, which explicitly steers the optimization trajectory toward hand regions. Extensive experiments validate the effectiveness of our method.

## Acknowledgements

This work was supported by the 2025 Tencent Rhino-bird Research Elite Program, the National Natural Science Foundation of China under Grant 62476133, and the Fundamental Research Funds for the Central Universities under Grant 11300-312200502507.

## Impact Statement

This work advances the field of human image animation, offering significant potential for applications in film production, virtual reality, and digital content creation. However, as with all high-fidelity generative technologies, there is a risk of misuse for creating misleading content. We advocate for the responsible development and deployment of such technologies, including the incorporation of watermarking and detection mechanisms to safeguard against malicious use. It is feasible to train a classifier to distinguish between real and generated videos based on their texture features.

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

# A. Appendix

## A.1. Ablation Study of Our Method and Supervised Fine-Tuning

To further validate the effectiveness of our proposed IPA, we conduct a comparative study against standard Supervised Fine-Tuning (SFT). For a fair comparison, the SFT model is fine-tuned using the exact same set of curated high-quality samples used in our IPA framework, employing the standard generative flow matching objective function (i.e., Eq. (2)) with our proposed Hand-Aware Local Optimization. The quantitative results on both the TikTok dataset and our proposed benchmark are reported in Tab. 6. We can observe that while SFT yields slight improvements in distribution-based metrics (e.g., FID-VID) compared to the pretrained baseline, it results in a significant degradation in pixel-wise metrics (e.g., PSNR). For instance, on the TikTok dataset, SFT drops SSIM from 0.777 to 0.715 and PSNR from 20.2 to 17.7. This phenomenon suggests that naively SFT by a small set of self-generated high-quality samples leads to severe overfitting and mode collapse, thereby harming the model's generalization ability. More importantly, this experiment provides even stronger evidence for the effectiveness of our proposed IPA, as merely relying on direct fine-tuning with carefully curated high-quality samples proves ineffective.

*Table 6.* Ablation study of our method and SFT.

| Dataset | Methods | FID-VID ↓ | FVD ↓ | SSIM ↑ | PSNR ↑ |
|---|---|---|---|---|---|
| | Baseline | 13.4 | 427 | 0.777 | 20.2 |
| TikTok Benchmark | Baseline+SFT | 12.8 | 391 | 0.715 | 17.7 |
| | Ours | **5.9** | **255** | **0.841** | **23.8** |
| | Baseline | 12.5 | 327 | 0.668 | 18.2 |
| Our Benchmark | Baseline+SFT | 11.8 | 328 | 0.666 | 16.9 |
| | Ours | **6.3** | **224** | **0.757** | **21.5** |

## A.2. Ablation Study of Regularized SFT

To isolate IPA's contribution, we implement a regularized SFT baseline: SFT (good samples) + HALO + LoRA + an L2 anchor regularizer ($\mathcal{L} = \mathcal{L}_{\text{SFT}} + ||v_\theta - v_{\text{ref}}||^2$). All other settings remain identical to our IPA run. The results on our benchmark are listed in Tab. 7. We can observe that while the regularizer mitigates SFT's catastrophic forgetting, its performance still vastly trails IPA. IPA succeeds because its dynamic objective severely penalizes only excessive prior deviations, outperforming static regularization.

*Table 7.* Ablation study of regularized SFT.

| Methods | FID-VID ↓ | FVD ↓ | SSIM ↑ | PSNR ↑ |
|---|---|---|---|---|
| Baseline | 12.5 | 327 | 0.668 | 18.2 |
| Baseline+SFT | 11.8 | 328 | 0.666 | 16.9 |
| Baseline+SFT+Regularizer | 11.2 | 308 | 0.673 | 18.5 |
| Ours | **6.3** | **224** | **0.757** | **21.5** |

## A.3. Comparison with KTO

To further demonstrate the effectiveness of IPA, we implement KTO (Li et al., 2024) as a relevant baseline, as KTO can use unpaired data as bad samples. For a fair comparison, we use the same 93 high-quality videos as good samples. We then randomly sample 93 unpaired videos as bad samples, and the base model and training steps are also identical for fine-tuning. The results on our benchmark are listed in Tab. 8. We can observe that IPA significantly outperforms KTO on our benchmark, further proving its superior effectiveness.

*Table 8.* Comparison with KTO.

| Methods | Requires Bad | FID-VID ↓ | FVD ↓ | SSIM ↑ | PSNR ↑ |
|---|---|---|---|---|---|
| Base Model | - | 12.5 | 327 | 0.668 | 18.2 |
| KTO (Li et al., 2024) | Yes | 10.9 | 291 | 0.689 | 19.1 |
| Ours | No | **6.3** | **224** | **0.757** | **21.5** |

## A.4. Empirical Observation of the Log-Sigmoid Saturation

In this section, we track the KL-divergence gap term and the total loss across the 1,000 training steps to explore the saturation mechanism during IPA training. As shown in Fig 5, we can conclude the following observations:

- **Initialization (Steps 0-100):** Initially, $v_\theta \approx v_{\text{ref}}$, meaning $\Delta \approx 0$. The loss evaluates to $-\log\sigma(0) \approx 0.69$, providing a strong initial gradient pulling the model towards the high-quality samples.

- **Active Learning (Steps 100-600):** As $v_\theta$ successfully approximates the hand structures, the term $\|v - v_\theta\|_2^2$ shrinks. Because $\|v - v_{\text{ref}}\|_2^2$ is a constant, $\Delta$ becomes increasingly positive.

- **Gradient Saturation (Steps 600-1000):** As $\Delta$ grows positive, the sigmoid output approaches 1.0. The loss term $-\log(1.0)$ approaches 0. The curve distinctly plateaus. This plateau empirically proves our saturation mechanism.

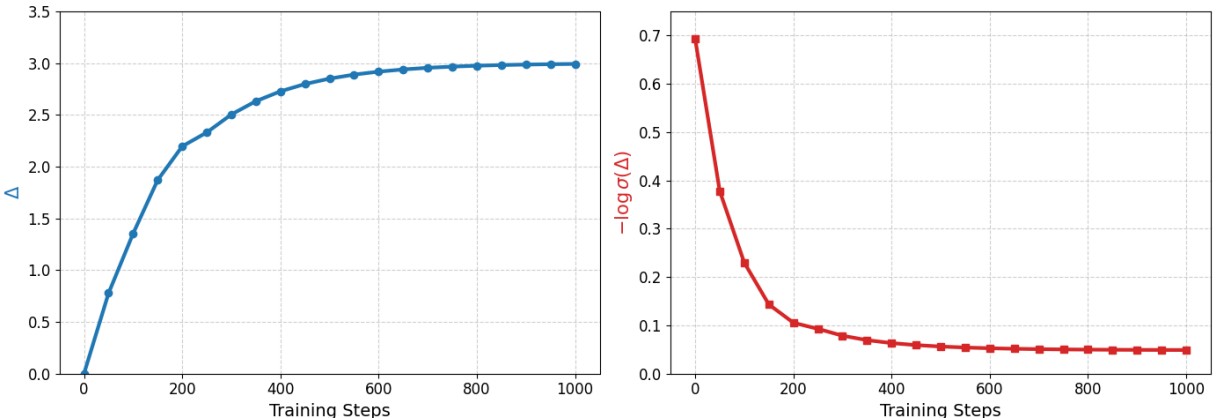

*Figure 5.* Empirical observation of $\Delta$ and $-\log\sigma(\Delta)$ during IPA training.

## A.5. Quantitative Results of Ablation Study on Different $\beta$

In this section, we provide the detailed quantitative results for the ablation study on the hyperparameter $\beta$, corresponding to the trends visualized in Fig. 4. Tab. 9 lists the performance metrics across a wide range of $\beta$ values from 200 to 2000 on both the TikTok benchmark and our proposed benchmark. The numerical data corroborates our analysis in Sec. 5.3.

## A.6. Qualitative Analysis of Ablation Study for Different $\beta$

We now visually analyze the results generated by models trained with varying $\beta$ to further investigate the effect of this hyperparameter. Fig. 6 visualizes the generated samples with $\beta = 200$, $\beta = 600$, and $\beta = 2000$. The following observations can be made: **i)** For $\beta = 200$, while hand quality is decent, the model produces anatomically impossible artifacts, i.e., an extraneous third hand. This demonstrates that an excessively small $\beta$ makes the model prone to overfitting, thereby degrading performance. **ii)** When $\beta = 2000$, the generated hands suffer from blurry artifacts and distortions. This indicates that an excessively large $\beta$ leads to model underfitting. **iii)** For $\beta = 600$, we observe that the generated hands exhibit clear structures and are devoid of anatomically impossible artifacts. This aligns with the optimal quantitative performance achieved at this setting.

## A.7. Human Preference Study

Following the MimicMotion evaluation protocol (Zhang et al., 2025), we conduct a Human Preference Study (10 evaluators) on 30 challenging videos. For baselines, we compare our method against three representative methods: MimicMotion, VACE, and Wan-Animate. For each case, evaluators are shown the ground truth pose and video, along with the video generated by our method and a baseline video (presented side-by-side in randomized order). Evaluators are asked to vote for the video that exhibited "more anatomically correct, stable, and artifact-free hand structures," with options for "Win" and "Lose". The summarized results are listed in Tab. 10, which shows the consistent and overwhelming preference for IPA, confirming its effectiveness.

*Table 9.* Ablation study on different $\beta$.

| Dataset | $\beta$ | FID-VID ↓ | FVD ↓ | SSIM ↑ | PSNR ↑ |
|---|---|---|---|---|---|
| TikTok Benchmark | 200 | 11.1 | 341 | 0.765 | 19.7 |
| | 400 | 8.6 | 287 | 0.824 | 22.8 |
| | 600 | **5.9** | **255** | **0.841** | **23.8** |
| | 800 | 5.9 | 257 | 0.835 | 23.7 |
| | 1000 | 6.9 | 278 | 0.832 | 23.2 |
| | 1200 | 7.6 | 272 | 0.827 | 23.0 |
| | 1400 | 7.4 | 283 | 0.817 | 22.6 |
| | 1600 | 8.6 | 299 | 0.807 | 22.2 |
| | 1800 | 8.8 | 296 | 0.803 | 21.9 |
| | 2000 | 9.5 | 313 | 0.793 | 21.5 |
| Our Benchmark | 200 | 11.7 | 334 | 0.696 | 18.0 |
| | 400 | 8.7 | 275 | 0.741 | 20.4 |
| | 600 | **6.3** | **224** | **0.757** | **21.5** |
| | 800 | 6.4 | 225 | 0.751 | 21.1 |
| | 1000 | 7.0 | 236 | 0.746 | 20.8 |
| | 1200 | 7.8 | 263 | 0.743 | 20.7 |
| | 1400 | 8.4 | 278 | 0.741 | 20.4 |
| | 1600 | 8.2 | 281 | 0.736 | 20.0 |
| | 1800 | 8.7 | 292 | 0.733 | 19.7 |
| | 2000 | 9.6 | 308 | 0.727 | 19.2 |

*Table 10.* Human Preference Study.

| Comparisons | Ours Win (%) | Ours Lose (%) |
|---|---|---|
| Ours vs. MimicMotion | 91.7 | 8.3 |
| Ours vs. VACE | 87.3 | 12.7 |
| Ours vs. Wan-Animate | 83.0 | 17.0 |

## A.8. More Visualization Results

In this section, we provide more comprehensive visual comparisons of different methods in Fig 7 and Fig 8 to demonstrate the effectiveness of our method. In addition, we provide more showcases of human image animation generated by our method in Fig 9 and Fig 10.

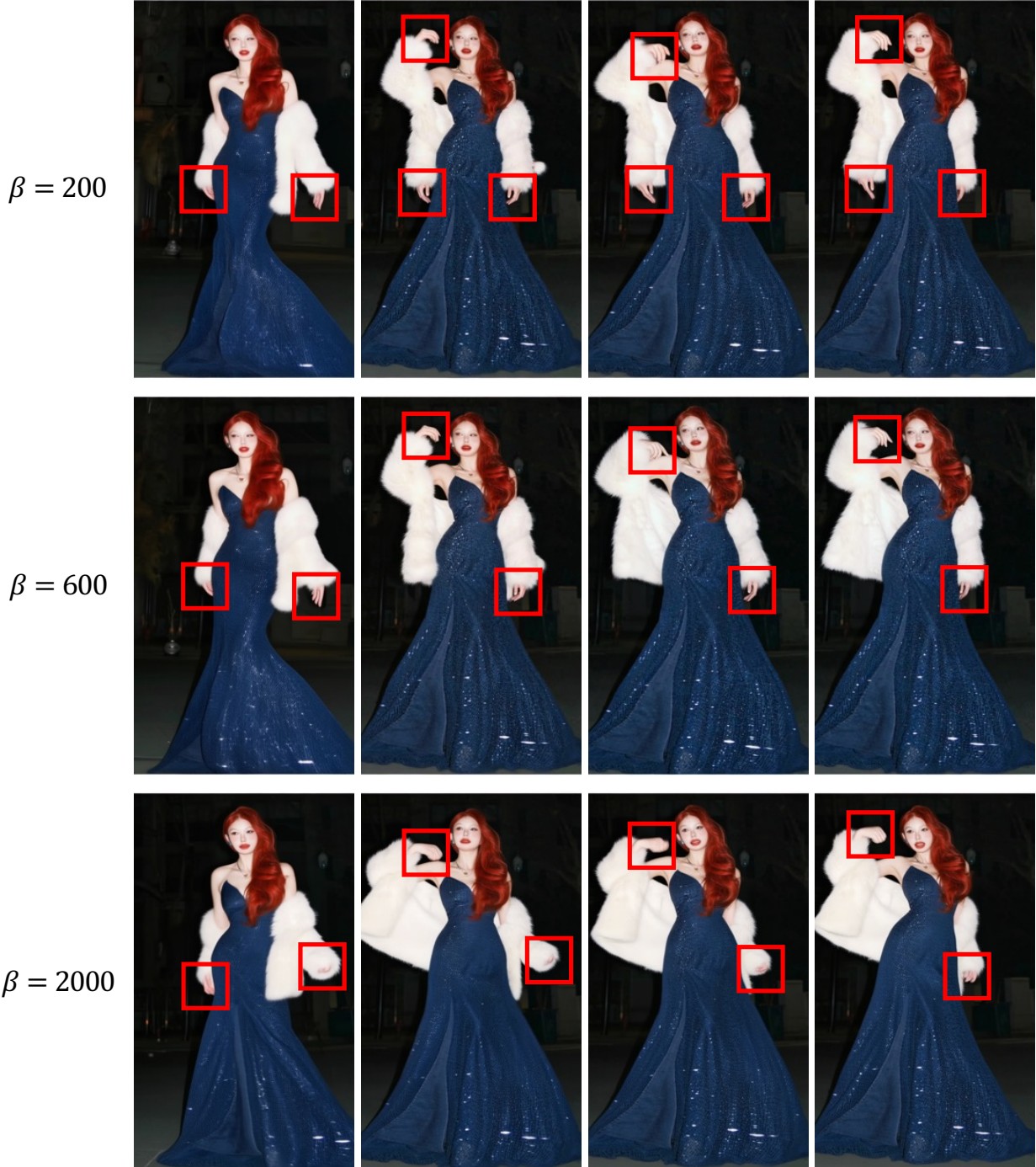

*Figure 6.* Visual results for different $\beta$. For $\beta = 200$, the model produces anatomically impossible artifacts (i.e., an extraneous third hand). When $\beta = 2000$, the generated hands suffer from blurry artifacts and distortions. For $\beta = 600$, the generated hands exhibit clear structures and are devoid of anatomically impossible artifacts.

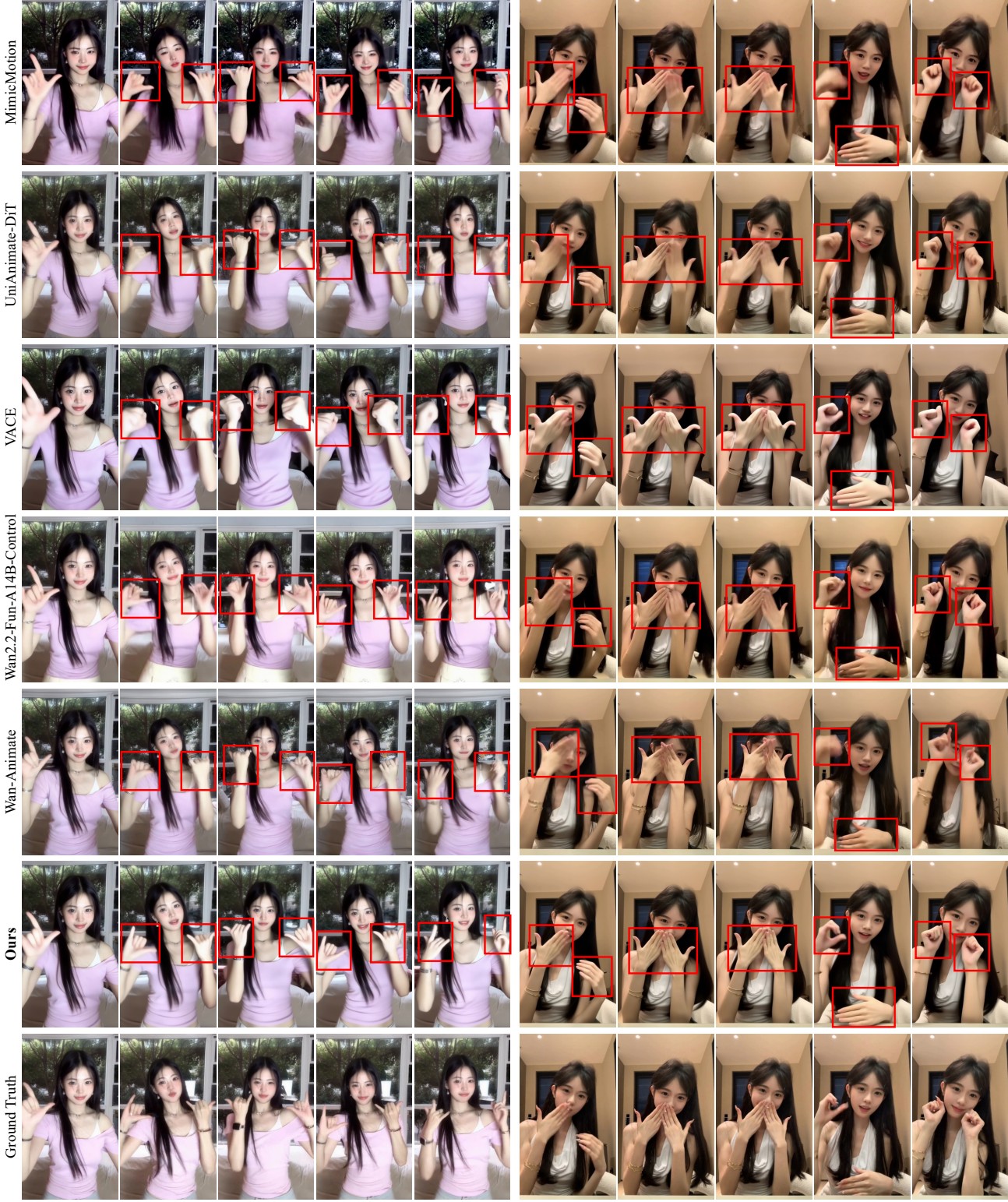

*Figure 7.* Complete visual comparisons of different methods for the case of Fig. 2.

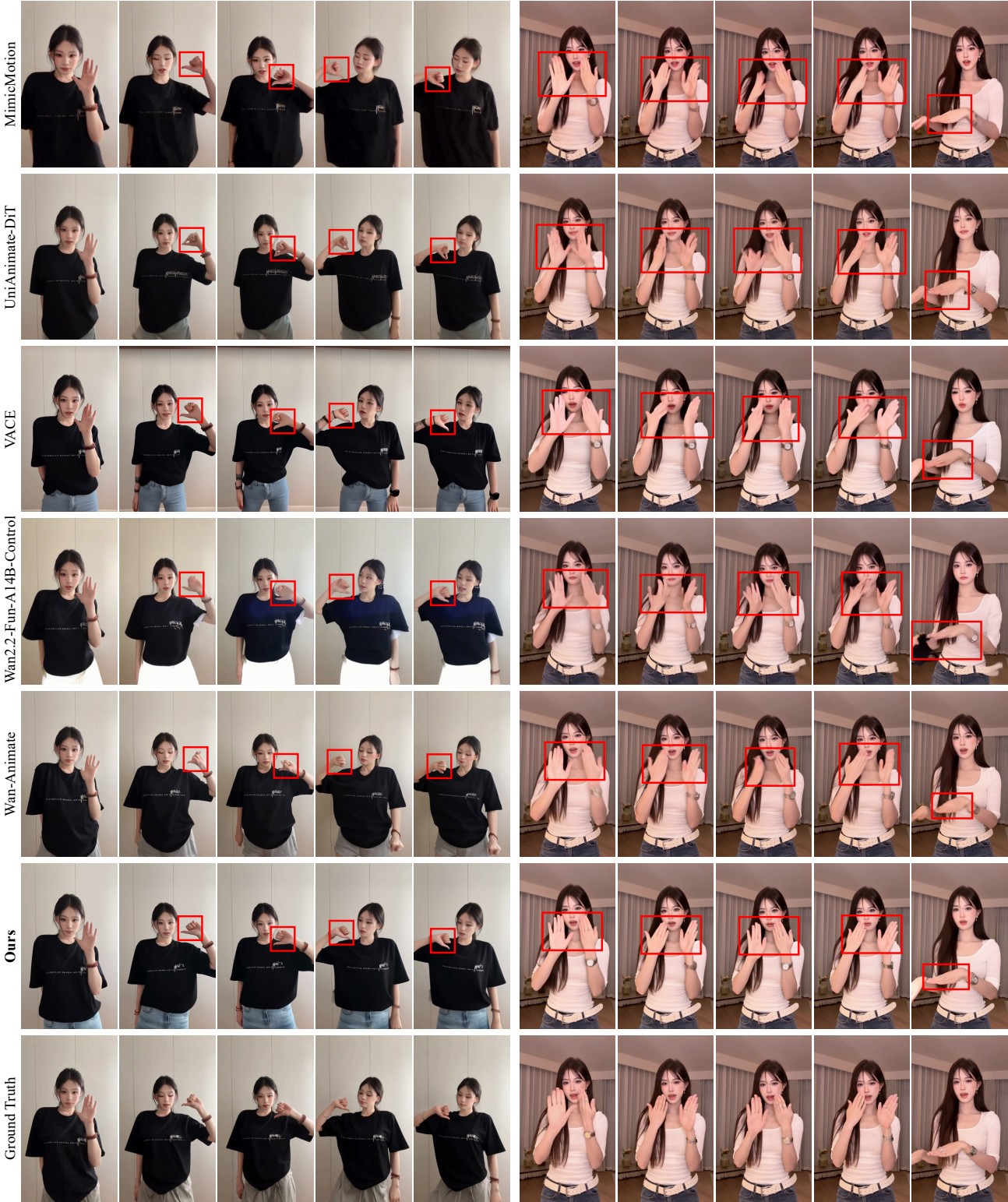

*Figure 8.* More visual comparisons of different methods.

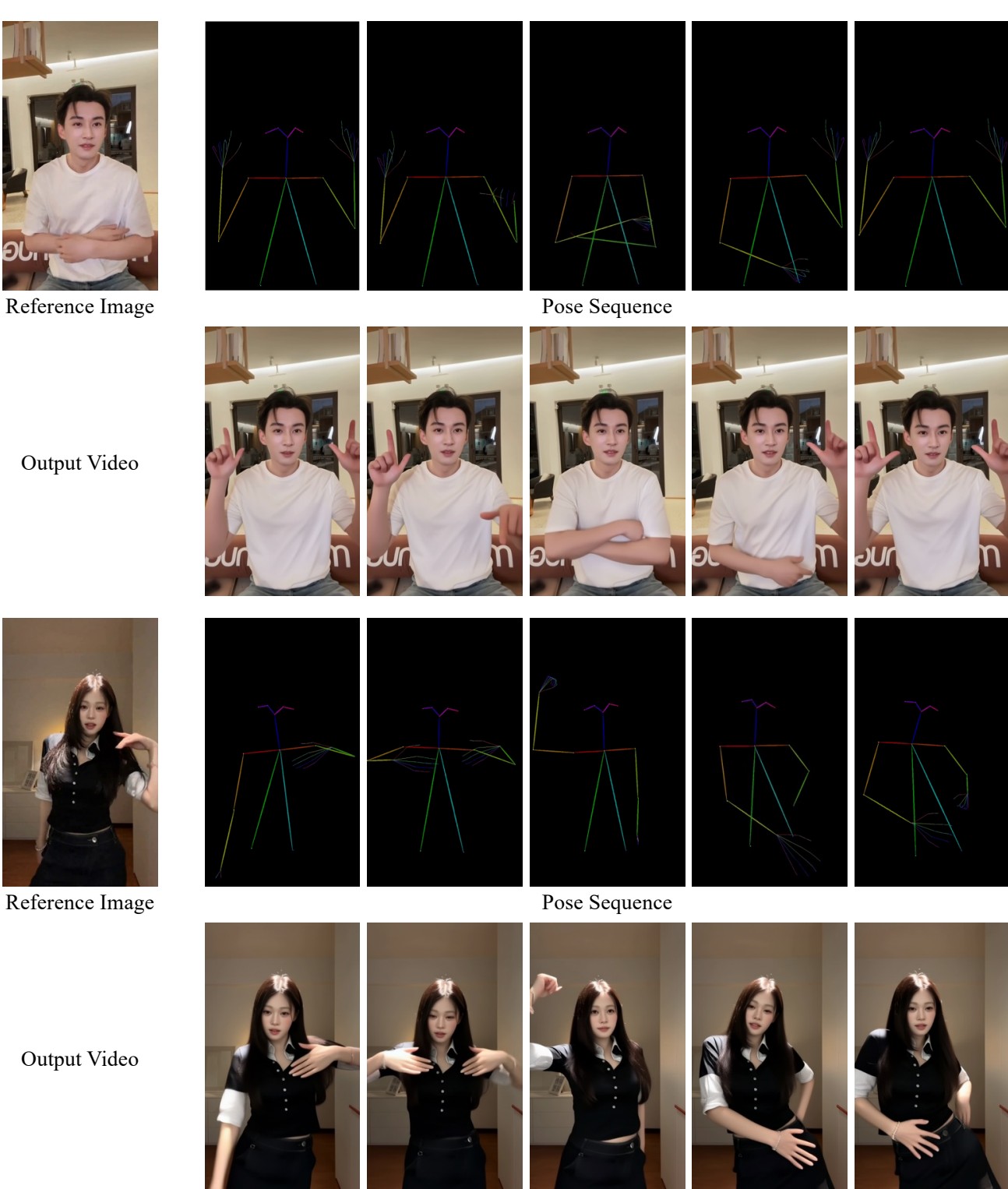

*Figure 9.* More showcases of human image animation generated by our method.

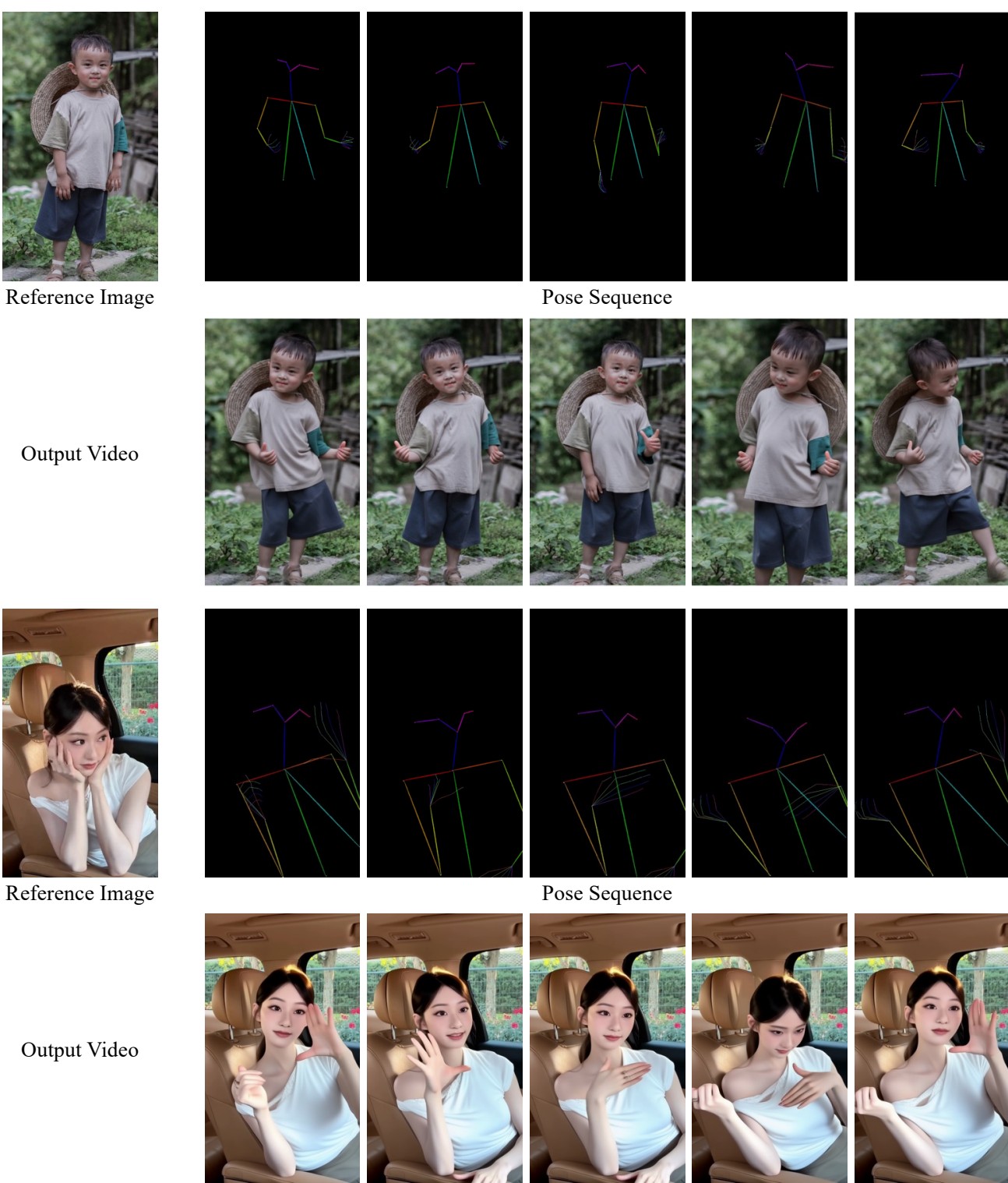

*Figure 10.* More showcases of human image animation generated by our method.

