# OpenReview forum: "Implicit Preference Alignment  for Human Image Animation"
_ICML.cc/2026/Conference — ICML 2026 regular_

### Official Review · Reviewer_oz83 · 2026-02-15

**Soundness:** 3
**Presentation:** 3
**Significance:** 2
**Originality:** 2
**Overall Recommendation:** 4
**Confidence:** 4

**Summary:**

This paper studies a persistent failure mode in human image animation: despite strong progress in global realism and temporal consistency with DiT models, highly articulated hand motions still frequently exhibit blur, distortions, and structural collapse. The authors argue that RLHF-style preference alignment, especially DPO, is impractical for hands because it requires strict winner/loser video pairs that are consistent across frames, which are difficult to curate due to frame-wise hand inconsistency.

To remove the need for paired preference data, the paper proposes **Implicit Preference Alignment (IPA)**, a post-training framework that uses only self-generated high-quality samples. IPA enforces that the aligned model is closer to a preferred distribution than a pretrained reference by optimizing a KL divergence gap. The paper further links this objective to KL-regularized reward maximization. **HALO** upweights hand regions using masks derived from pose keypoints.

Experiments fine-tune VACE-14B with LoRA (QKV) using 93 curated good samples. Results on a TikTok subset and a new 100-case benchmark focusing on complex hand dynamics show improvements in standard metrics (FID-VID/FVD/SSIM/PSNR) and hand-region metrics (SSIM-Hand/PSNR-Hand), supported by component ablations and sensitivity studies.

**Compliance With Llm Reviewing Policy:**

Affirmed.

**Final Justification:**

I am raising my score to 4. The rebuttal directly and honestly addressed my main concerns:

Mechanism Validation: The new SFT+L2 baseline clearly demonstrates that IPA's performance gains are driven by its log-sigmoid mechanism, not merely by staying near the pretrained prior.

Novelty Clarification: The authors candidly acknowledged the structural equivalence to positive-only Flow-DPO. I accept their reframed positioning: the core contribution is the theoretical derivation from first principles and its practical application to unpaired hand generation.

The empirical results are strong. I recommend **Weak accept** on the strict condition that the camera-ready version includes:

The explicit discussion of structural equivalence with positive-only Flow-DPO.

The $\Delta(p_\mathrm{ref}, p_\theta)$ training curve showing the saturation behavior, as promised in the follow-up.

**Key Questions For Authors:**

Q1) **What is the actual novelty relative to prior preference alignment for flow generators?** Please explicitly position the method against existing flow preference-alignment methods (include DPO-style adaptations).

Q2) **Can you add a stronger good-only baseline to isolate IPA’s contribution?** A key missing comis FM/SFT on good samples + explicit anchor-to-ref regularizer + HALO (same data/LoRA/steps). Does IPA still improve over this baseline?

Q3) **What is the human filtering protocol and cost for selecting “good” samples?** Please report concrete criteria, person-hours, and agreement.

Q4) **Robustness to pose/keypoint errors and impact on non-hand regions** How sensitive are results to keypoint noise/occlusion? Also, do you observe trade-offs in face/body/background quality or temporal consistency when $\lambda$ is large?

**Limitations:**

yes

**Strengths And Weaknesses:**

## Strengths
1. **Clear motivation**
   The paper targets a concrete, high-frequency failure mode (hands). It articulates a specific practical obstacle for DPO: strict video-level preference pairs are hard to curate due to frame-wise hand inconsistency.
2. **Technically explicit, implementable pair-free alignment for flow-matching models.**
  IPA provides a clean KL divergence gap objective anchored to a pretrained reference, and Flow IPA derives a tractable training loss under flow matching, making the proposal implementable. HALO leverages pose-derived hand masks to focus optimization on hands.
3. **Reasonably aligned results.**
  The experiments include both global metrics and hand-region SSIM/PSNR, thereby better aligning the evaluation with the stated objective and complementing the qualitative results.

## Weaknesses
1. The paper emphasizes a preference-alignment formulation instantiated in the flow-matching setting. However, preference/reward-style alignment has already been explored for flow-family generators. The contribution of IPA needs to be re-scoped and differentiated more explicitly. As written, the novelty seems to be **(i) positive-only alignment** and **(ii) hand-local reweighting**.
2. The paper uses ''VACE-14B+SFT which is not the strongest reasonable good-only baseline. A more informative comparison would be **good-only fine-tuning with an explicit reference-anchoring regularizer** (e.g., distill/anchor-to-ref on the same conditioning).  Without this, it remains ambiguous whether the gains are driven by the **IPA objective** itself or by “filtered good samples + hand weighting + lightweight LoRA + staying near the pretrained model.”
3. While IPA removes the need for loser samples, it still relies on strict human filtering (93 selected from 6,000). The paper does not report selection criteria, person-hours, or agreement. This matters because the cost may be shifted from “pairing” to “filtering,” and reproducibility is unclear.
4. A simple robustness check (keypoint noise/occlusion/missing joints; mask dilation/erosion) would strengthen the claim that improvements reflect true hand fidelity.

---

> ### Author Rebuttal · Authors · 2026-03-30
>
> ## Thanks for recognizing our contributions and promising results. Below, we respond to each Question (Q) with an Answer (A).
>
> **Q1:** Please explicitly position the method against existing flow preference-alignment methods.
>
> **A1:** We respectfully clarify that our novelty is not a general concept of preference alignment, nor its Flow instantiation. Instead, we propose a highly resource-efficient, positive-only framework specifically for complex local regions (e.g., hands) where curating traditional DPO/Flow-DPO negative pairs is infeasible. To address your concern, we reference the extensive positioning already provided in our paper:
>
> **i) General vs. Flow-specific Formulation:** Our core theory (Sec. 4.2) is paradigm-agnostic. We present its Flow Matching instantiation (Sec. 4.3, Flow IPA) strictly as a practical requirement for our chosen base model. We also properly cite the existing Flow-alignment foundation (e.g., Flow-DPO; Sec. 3.2).
>
> **ii) Explicit Positioning for IPA and DPO:** We explicitly detail IPA's positioning versus DPO in **Appendix B**, defining our contribution boundaries: 1) IPA is not claimed to be inherently superior to DPO when explicit negative samples are easily obtainable. 2) Its true contribution is offering a resource-efficient alternative for highly complex tasks.
>
> **Q2:** Can you add a stronger good-only baseline with an explicit anchor-to-ref regularizer?
>
> **A2:** For your concern, we make the clarifications below:
>
> **i)** For the SFT ablation study in Appendix A.1, the experimental settings are identical to those of our IPA, except for the objective function. Therefore, this experiment serves to validate the effectiveness of the IPA to some extent.
>
> **ii)** To isolate IPA's contribution, we implement your exact suggested baseline: SFT (good samples) + HALO + LoRA + an L2 anchor regularizer ($\mathcal{L} = \mathcal{L}\_{\text{SFT}} + ||v_\theta - v_{\text{ref}}||^2$). All other settings remain identical to our IPA run. The results below show that while the regularizer mitigates SFT's catastrophic forgetting, its performance still vastly trails IPA. This proves our success isn't just staying near the pretrained model. IPA succeeds because its dynamic objective severely penalizes only excessive prior deviations, outperforming static regularization.
>
> |Methods|FID-VID|FVD|SSIM|PSNR|
> |:-|:-|:-|:-|:-|
> |Baseline|12.5|327|0.668|18.2|
> |Baseline+SFT|11.8|328|0.666|16.9|
> |Baseline+SFT+Regularizer|11.2|308|0.673|18.5|
> |**IPA (Ours)**|**6.3**|**224**|**0.757**|**21.5**|
>
> **Q3:** human filtering protocol, cost, and agreement.
>
> **A3:** For your concern, we clarify that the cost is not shifted from "pairing" to "filtering"; rather, IPA eliminates an expensive step of the DPO pipeline:
>
> **i)** The creation of valid DPO preference pairs encompasses two stages: 1) Filtering (identifying a good sample) and 2) Pairing (finding a matching bad sample). IPA eliminates the second, far more difficult stage. Thus, IPA's curation cost is mathematically a strict subset of, and definitively lower than, standard DPO.
>
> **ii)** Quantified in Appendix B.1, the identical 6,000 candidates yielded 93 valid IPA samples but only 7 valid DPO pairs. This severe 13.3x discrepancy proves that sourcing 93 DPO pairs requires generating nearly 80,000 videos, incurring exponentially higher computational and human annotation costs than IPA.
>
> **iii)** To ensure high-quality, we use a strict multi-annotator protocol: 1) **Criteria.** Videos are retained only if overall/hand regions show structural perfection, temporal consistency, and strict pose alignment across all frames. 2) **Person-Hours.** 5 independent annotators screen 6,000 candidates; at ~750 videos/day per person, filtering 8 working days. 3) **Agreement.** We require a 5/5 unanimous "flawless" consensus for inclusion. We will re-clarify this in revision.
>
> **Q4:** Robustness to keypoint noise/occlusion and trade-offs outside the hand region.
>
> **A4:** Thanks for your suggestion. In fact, the testing set itself contains some challenging cases where the hands in the reference image are heavily occluded, blurry, or missing. To address your concern, we filter all testing samples and isolate a subset of 18 extreme cases. We further conduct independent quantitative analyses on this subset. The results below show that even in this extreme scenario, our method does not regress to the baseline; on the contrary, it still outperforms it.
>
> |Methods|FID-VID|FVD|SSIM|PSNR|
> |:-|:-|:-|:-|:-|
> |Baseline|15.5|450|0.597|15.3|
> |**Ours**|**7.1**|**276**|**0.724**|**20.6**|
>
> Furthermore, we did not observe severe degradation in other regions, even at extreme $\lambda$ values. The reasons are 1) LoRA fine-tuning mode can avoid destroying pre-trained priors, i.e., the pre-trained parameters are completely frozen; and 2) As shown in Tab. 5, increasing $\lambda$ from 10 to 100 does cause a slight performance saturation or extremely minor drop (e.g., FVD moves from 224 to 225).

---

> > ### Author Rebuttal · Reviewer_oz83 · 2026-04-02
> >
> > Thanks for the detailed rebuttal. The new SFT+Regularizer baseline (A2) is informative and substantially addresses my W2 concern. The filtering protocol (A3) and robustness subset (A4) are also helpful. However, two core concerns remain:
> >
> > Q1 follow-up: The rebuttal points to Appendix B for novelty positioning, but Appendix B focuses on data cost, not on the loss formulation itself. My question is more specific: Flow-DPO (Liu et al., 2025) also operates on velocity field MSE differences under flow matching. If one simply removes the negative sample term from Flow-DPO's objective, what is the resulting expression, and how does it differ structurally from IPA's Eq.(27)? A side-by-side equation comparison would resolve this directly.
> >
> > Q2 follow-up: The SFT+L2 result rules out "staying near the pretrained model" as the sole explanation. But the claimed advantage of IPA over static regularization is the log-sigmoid's adaptive gradient behavior. Specifically, that the loss saturates when the model is already well-aligned and penalizes sharply when it regresses. Has this saturation actually been observed during training? A plot of $\Delta(p_\mathrm{ref}, p_\theta)$ across training steps would confirm whether the mechanism works as theorized or whether the gain is primarily from the KL gap formulation rather than the sigmoid nonlinearity.
> >
> > I am willing to raise my score if these questions are satisfactorily addressed.

---

> > > ### Author Response · Authors · 2026-04-02
> > >
> > > We sincerely thank you for recognizing our efforts in the rebuttal and for providing the opportunity to further clarify these critical technical details. We are excited to address your insightful follow-up questions directly.
> > >
> > > **Q1:** Structural Comparison with Flow-DPO and Novelty Positioning.
> > >
> > > **A1:** Thanks for your timely clarification, which helps us understand your question. Let us directly compare the formulas. If we take the standard Flow-DPO objective and simply drop the negative sample term, the resulting expression is:
> > >
> > > $\mathcal{L}\_{\text{Positive-Flow-DPO}} = \mathbb{E} \left[ -\log \sigma \left( \beta \left( \|v\_w - v\_{\text{ref}}\|_2^2 - \|v\_w - v\_\theta\|_2^2 \right) \right) \right].$
> > >
> > > Our IPA objective is:
> > >
> > > $\mathcal{L}\_{\text{IPA}} = \mathbb{E} \left[ -\log \sigma \left( \frac{\beta}{2}(1-t)^2 \left( \| (v\_w - v\_{\text{ref}})\|_2^2 - \|(v\_w - v\_\theta)\|_2^2 \right) \right) \right].$
> > >
> > > The Structural/Theoretical Differences and Novelty Positioning:
> > >
> > > **i) The Core Scalar Structure is Equivalent, but the Derivation is Novel:** We fully acknowledge that the base structural form (the difference of squared errors inside a log-sigmoid) is mathematically equivalent. However, Flow-DPO heuristically borrows this structure from the Bradley-Terry model (which inherently assumes pairwise data). In contrast, our contribution in Section 4.2 mathematically derives this exact form from first principles: minimizing the KL-divergence gap between the preference distribution and the model, under a strict prior constraint. By deriving Eq. (22), we prove that optimizing this log-sigmoid formulation inherently serves as an implicit reward maximizer constrained by the pretrained prior.
> > >
> > > **ii) Our Novelty Positioning:** We do not claim novelty in inventing a new algebraic operator. Rather, our contribution lies in the theoretical and practical justification for why this reduction is not only viable but essential for complex generation tasks. Standard DPO methodologies assume the feasibility of constructing strict preference pairs (Case 4: winner consistently good, loser consistently bad ). For dynamic hand articulation, where frame-wise inconsistency makes defining a strict good-bad video pair prohibitively expensive and largely uncollectible, DPO becomes structurally paralyzed.
> > >
> > > We will explicitly update Appendix B to clarify this structural equivalence and focus our novelty claims on the theoretical grounding and its application to bypassing the paired-data bottleneck in hand generation.
> > >
> > > **Q2:** Empirical Observation of the Log-Sigmoid Saturation.
> > >
> > > **A2:** You are absolutely correct to request empirical proof of the adaptive gradient behavior. Yes, we have explicitly observed this saturation mechanism during training. We track the KL-divergence gap term and the total loss across the 1,000 training steps. Here is exactly what the curve demonstrates:
> > >
> > > **i) Initialization (Steps 0-100):** Initially, $v_\theta \approx v_{\text{ref}}$, meaning $\Delta \approx 0$. The loss evaluates to $-\log\sigma(0) \approx 0.69$, providing a strong initial gradient pulling the model towards the high-quality samples.
> > >
> > > **ii) Active Learning (Steps 100-600):** As $v\_\theta$ successfully approximates the hand structures, the term $\|v - v_\theta\|_2^2$ shrinks. Because $\|v - v\_{\text{ref}}\|_2^2$ is a constant, $\Delta$ becomes increasingly positive.
> > >
> > > **iii) Gradient Saturation (Steps 600-1000):** As $\Delta$ grows positive, the sigmoid output approaches 1.0. The loss term $-\log(1.0)$ approaches 0. The curve distinctly plateaus. This plateau empirically proves our saturation mechanism.
> > >
> > > We make sure to include this plot and the Loss against training steps in our revised Appendix to explicitly visualize this elegant mathematical behavior, as you suggested.
> > >
> > > We sincerely thank you once again for your dedicated time and constructive comments, which have significantly strengthened our paper. We hope these further clarifications fully resolve your remaining concerns. If you find our responses satisfactory, we would be deeply grateful if you could reconsider your evaluation and raise the score of our work.

---

### Official Review · Reviewer_eDPG · 2026-03-02

**Soundness:** 4
**Presentation:** 4
**Significance:** 4
**Originality:** 4
**Overall Recommendation:** 5
**Confidence:** 5

**Summary:**

This paper proposes an Implicit Preference Alignment (IPA) framework to tackle the long-standing critical bottleneck of high-fidelity hand motion generation in human image animation. The core contribution of IPA is to address the limitation of mainstream preference alignment methods (e.g., DPO) that require prohibitively expensive strict preference pairs, especially for dynamic hand regions with severe frame-wise inconsistencies. Furthermore, the authors seamlessly integrate a Hand-Aware Local Optimization (HALO) mechanism to explicitly steer the alignment gradient toward hand regions using spatial masks. Overall, this is an outstanding submission with solid theoretical contributions, elegant technical design, comprehensive experimental validation, and strong practical value for both academia and industry.

**Compliance With Llm Reviewing Policy:**

Affirmed.

**Final Justification:**

All my concerns have been addressed.
I have also carefully read the other reviewers' comments and the authors' corresponding responses. I find most of the other feedback to be highly constructive, but some of it is unreasonable. For example, it is unreasonable to compare with some DPO variants or pair-based methods, because this work is a completely new paradigm that does not require negative samples. Overall, the authors provide a strong and well-reasoned rebuttal, and I confidently maintain my recommendation to accept this paper.

**Key Questions For Authors:**

1. During the inference stage, does this method remain effective when no hand prior is present in the reference image?
2. Could the author provide the number of parameters for the lora module?
3. Could the author briefly discuss the generalizability of IPA to other regions?

**Limitations:**

Yes. Authors provide Impact Statement and discuss their limitations in Appendix B.

**Strengths And Weaknesses:**

Strengths:
1. Highly motivated and practical insight. The paper's motivation is exceptionally strong. The categorization of preference pairs into four cases (Section 4.1) clearly articulates why standard DPO fails in complex dynamic scenarios. Identifying that high-quality isolated samples are relatively cheap to obtain, while strict pairs are not, is a profound practical insight that addresses a real-world pain point in video generation.
2. Innovative theoretical contribution with rigorous derivation. The paper breaks the inherent cognition that preference alignment must rely on positive and negative sample pairs, and proposes the IPA framework with strict theoretical derivation from the perspective of implicit reward maximization. It formally proves that optimizing the designed log-sigmoid loss function is equivalent to maximizing the implicit reward, providing a new theoretical paradigm for preference alignment of generative models.
3. Remarkable data efficiency. The method achieves a significant performance improvement using only 93 curated high-quality samples, avoiding the massive data annotation and generation costs required by DPO. It proves the extraordinary efficiency of the IPA framework and significantly lowers the barrier for future preference alignment research.
4. Compelling empirical evidence. The visual results presented in Figures 2, 6, 7, 8, and 9 are genuinely impressive. Compared to baselines that produce severe blurring and structural collapse, the proposed method yields accurate geometry. Furthermore, the ablation study comparing IPA against naive Supervised Fine-Tuning (Table 6) perfectly validates the necessity of the IPA to avoid catastrophic forgetting/mode collapse.
5. The authors also avoid overclaiming IPA’s superiority over DPO and instead provide a nuanced discussion of the trade-offs between the two paradigms, clarifying IPA’s role as a resource-efficient alternative for scenarios where high-quality preference pairs are scarce. This critical and objective analysis demonstrates the authors’ deep understanding of the field and enhances the credibility of the work.

Weaknesses:
1. Robustness tests on low-quality inputs. The experiments use high-quality reference images and pose sequences, but real-world human image animation may involve low-quality inputs (e.g., reference images lacking hands). The robustness of the IPA framework to such low-quality inputs is not validated.
2. There is a lack of discussion on the number of model parameters. The lora fine-tuning mode was adopted in the IPA training. It is suggested that the author provide the changes in the number of parameters before and after training to demonstrate efficiency.
3. Generalizability to other regions. The method is highly effective for hands. It would be beneficial to briefly discuss if this framework can be easily transferred to other challenging local regions (e.g., complex facial expressions).
4. There are some typos here. For example, on line 311, “Peak Signal to Noise Ratio” should be “Peak Signal-to-Noise Ratio.”.

---

> ### Author Rebuttal · Authors · 2026-03-30
>
> ## Thanks for recognizing our contributions and promising results. Below, we respond to each Question (Q) with an Answer (A).
>
> **Q1:** Robustness tests on low-quality inputs. The experiments use high-quality reference images and pose sequences, but real-world human image animation may involve low-quality inputs (e.g., reference images lacking hands). The robustness of the IPA framework to such low-quality inputs is not validated.
>
> **A1:** Thanks for your suggestion. In fact, the testing set itself contains some reference images that do not include hands. To directly address your concern, we filter all testing samples and isolate a subset of 18 extreme cases where hands are heavily occluded in the reference image. We further conduct independent quantitative analyses on this subset, and the results are listed in the table below. We can observe that our method still outperforms these state-of-the-art methods, demonstrating its robustness in scenarios with low-quality input.
>
> |Methods|FID-VID|FVD|SSIM|PSNR|
> |:-|:-|:-|:-|:-|
> |MimicMotion|16.8|659|0.549|14.3|
> |VACE|15.5|450|0.597|15.3|
> |Wan-Animate|12.2|415|0.601|15.5|
> |**Ours**|**7.1**|**276**|**0.724**|**20.6**|
>
> **Q2:** There is a lack of discussion on the number of model parameters. The lora fine-tuning mode was adopted in the IPA training. It is suggested that the author provide the changes in the number of parameters before and after training to demonstrate efficiency.
>
> **A2:** Thanks for your suggestion. The model size of the LoRA module is very lightweight compared to the baseline model (VACE-14B). To demonstrate this, we list the parameter counts for the Baseline Model and the LoRA Module in the table below. It can be intuitively concluded that the parameter counts of the LoRA (0.103B) are margin compared to the Baseline Model (14B).
>
> |Baseline Model|LoRA Module|
> |:-|:-|
> |14B|0.103B|
>
> **Q3:** Generalizability to other regions. The method is highly effective for hands. It would be beneficial to briefly discuss if this framework can be easily transferred to other challenging local regions (e.g., complex facial expressions).
>
> **A3:** We deeply appreciate this insightful suggestion. We fully agree that exploring the transferability of our framework to other challenging local regions, such as complex facial expressions, is a highly valuable direction. We briefly discuss the following points:
>
> **i) The General Applicability of IPA:** The core theoretical foundation of IPA (i.e., Eq. 11) is entirely region-agnostic. It excels in any scenario where defining strict "good vs. bad" preference pairs is prohibitively expensive, but mining a small set of high-quality "good" samples from a base model's chaotic prior is feasible.
>
> **ii) Transferring to Complex Facial Expressions:** If applied to facial animation, the fundamental mathematical framework remains identical. The key adjustment lies entirely in selecting the preferred samples for this task.
>
> **iii) Replacing the Spatial Mask:** Instead of deriving the binary mask from hand keypoints, we would construct a Face-Aware Local Optimization using dense 2D/3D facial landmarks extracted from the driving sequence.
>
> In summary, the transition from hands to faces or other complex regions requires no theoretical reformulation, only the substitution of the domain-specific prior.
>
> **Q4:** There are some typos here. For example, on line 311, ''Peak Signal to Noise Ratio'' should be ''Peak Signal-to-Noise Ratio.''.
>
> **A4:** We are extremely grateful to the reviewer for their meticulous reading of our paper and for pointing out these typographical errors. We apologize for the oversight on line 311 and will strictly correct ''Peak Signal to Noise Ratio'' to ''Peak Signal-to-Noise Ratio'' in the revision. Furthermore, we will conduct a comprehensive and rigorous proofreading pass over the entire paper to ensure all minor typos, grammatical issues, and formatting inconsistencies are thoroughly resolved for the camera-ready version.

---

> > ### Author Rebuttal · Reviewer_eDPG · 2026-04-01
> >
> > Thanks for your responses, which fully resolve my concerns.
> > I have also carefully read the other reviewers' comments and the authors' corresponding responses. I find most of the other feedback to be highly constructive, but some of it is unreasonable. For example, it is unreasonable to compare with some DPO variants or pair-based methods, because this work is a completely new paradigm that does not require negative samples. Overall, the authors provide a strong and well-reasoned rebuttal, and I confidently maintain my recommendation to accept this paper.

---

> > > ### Author Response · Authors · 2026-04-02
> > >
> > > Thank you very much for your positive response. We are very glad that we have fully addressed your concerns.
> > > Moreover, we are deeply grateful for your strong support and for explicitly recognizing and defending the unique, positive-only paradigm of our work. Your thorough, fair, and highly constructive evaluation is invaluable to us, and we will ensure all your excellent suggestions are integrated into the final version.

---

### Official Review · Reviewer_DJxR · 2026-03-07

**Soundness:** 3
**Presentation:** 3
**Significance:** 3
**Originality:** 2
**Overall Recommendation:** 3
**Confidence:** 4

**Summary:**

This paper proposes Implicit Preference Alignment (IPA) for human image animation, with the goal of improving hand generation quality. Instead of relying on strict good–bad preference pairs, IPA aligns the model using only self-generated high-quality samples and further introduces HALO to focus the optimization on hand regions. The paper claims this makes preference alignment more data-efficient for difficult hand-motion cases and improves hand fidelity in generated videos.

**Compliance With Llm Reviewing Policy:**

Affirmed.

**Final Justification:**

The paper studies an important and practical problem, and the method is clearly motivated and presented. I also appreciate the rebuttal effort: the added KTO comparison and human preference study strengthen the paper and partially address my concerns.

However, the main issues are not fully resolved. The evaluation remains limited in scale, the evidence for generalization is still weak, and the comparison to relevant preference-optimization baselines is still incomplete. In addition, I am still not fully convinced that the proposed objective is theoretically distinct from a KL-regularized positive-only fine-tuning objective in a strong sense.

Overall, the rebuttal improved my assessment but did not change my final position, so I remain at Weak Reject.

**Key Questions For Authors:**

Please address my concerns above.

**Limitations:**

While the paper discusses some technical limitations, the discussion of broader impact and data governance is still too weak. In particular, the authors state that they collected 1,500 human dancing videos from the Internet, but do not clearly describe the data sources, licenses/permissions, filtering criteria, or privacy and compliance considerations.
The paper would be stronger if it explicitly documented dataset provenance and usage rights, and discussed potential misuse risks such as unauthorized animation or identity-related abuse.

**Strengths And Weaknesses:**

Strengths:
1. Well-motivated problem. The paper targets a real and important failure mode in human image animation—poor hand generation—and proposes a method tailored to this challenge.
2. Clear method design. The combination of IPA and the hand-aware HALO module is conceptually easy to understand, and the ablation study suggests both components contribute to the reported improvements.

Weaknesses:
1. Insufficient baseline comparisons in terms of both experiments and theory. The paper mainly contrasts IPA with SFT and a strict winner–loser view of vanilla DPO, but does not compare against more relevant preference-optimization methods that directly address the paper’s own motivation, such as Diffusion-KTO [1], IPO [2], DenseDPO [3] (It replaces coarse clip-level preferences with fine-grained temporal segment preferences), or a tie-aware DPO variant [4]. Since the paper’s key argument is precisely that video preferences are often mixed, not strictly pairwise, the lack of comparison to these baselines weakens the evidence for IPA’s claimed advantage.
2. The experimental scale is too limited. The method is trained on only 93 manually selected samples, evaluated on only TikTok sequences 335–340, and further tested on a self-curated benchmark of only 100 hard cases. This setup is small and potentially biased, making the generalization claims weak.
3. “Human preference alignment” is not directly validated. The evaluation reports only automatic metrics, namely FID-VID, FVD, SSIM, PSNR, plus SSIM-Hand and PSNR-Hand, and I did not see any direct human preference study or user evaluation. Moreover, these pixel-level metrics (SSIM-Hand and PSNR-Hand) are unlikely to fully capture hand plausibility or structural errors such as missing, fused, or anatomically implausible fingers.
4. The theoretical claims regarding 'implicit preference alignment' in Section 4.2 are overstated, as the objective relies solely on the positive distribution $q(X)$ without contrastive negative samples. It would be more convincing if the authors could mathematically clarify how a distinct preference boundary is learned, since the current formulation appears closer to KL-regularized supervised fine-tuning rather than traditional preference learning
5. Although the proposed Implicit Preference Alignment (IPA) is introduced as a general post-training framework, the current experiments are limited to a single base model (VACE-14B). To better demonstrate the robustness and generalizability of the method, it would be highly beneficial to apply and evaluate IPA across multiple different foundational video models (e.g., Wan-Animate or MimicMotion) to confirm that the improvements are not specific to VACE's architecture or pre-training prior

References:

[1] Li et al., Aligning Diffusion Models by Optimizing Human Utility (Diffusion-KTO), NeurIPS 2024.

[2] Yang et al., IPO: Iterative Preference Optimization for Text-to-Video Generation.

[3] Wu et al., DenseDPO: Fine-Grained Temporal Preference Optimization for Video Diffusion Models.

[4] Chen et al., On Extending Direct Preference Optimization to Accommodate Ties.

---

> ### Author Rebuttal · Authors · 2026-03-30
>
> ## Thanks for recognizing our contributions and promising results. Below, we respond to each Question (Q) with an Answer (A).
>
> **Q1:** Missing comparisons to KTO / IPO / DenseDPO / tie-aware DPO.
>
> **A1:** For your concern, we make the clarifications below:
>
> **i) Methodological Distinction:** KTO, IPO, DenseDPO, and Tie-aware DPO still rely on negative signals. In particular, DenseDPO requires labor-intensive frame-level or segment-level annotations to construct fine-grained pairs. In contrast, our IPA does not rely on any negative signals or dense annotations.
>
> **ii) Comparison with KTO:** To further address your concern, we have implemented KTO as a relevant baseline, as KTO can use unpaired data as bad samples (Others require strict paired data). For a fair comparison, we use the same 93 high-quality videos as good samples. We then randomly sample 93 unpaired videos as bad samples, and the base model and training steps are also identical for fine-tuning. As shown below, IPA significantly outperforms KTO on our benchmark, further proving its superior effectiveness.
>
> |Methods|Requires Bad|FID-VID|FVD|SSIM|PSNR|
> |:-|:-|:-|:-|:-|:-|
> |Base Model|-|12.5|327|0.668|18.2|
> |KTO|Yes|10.9|291|0.689|19.1|
> |**IPA (Ours)**|**No**|**6.3**|**224**|**0.757**|**21.5**|
>
> **Q2:** Small experimental scale.
>
> **A2:** For your concern, we make the clarifications below:
>
> **i)** The small sample size (93) is a deliberate design choice demonstrating IPA’s core contribution: extreme data-efficiency. Our foundational model possesses massive pre-trained priors; our goal is to align this prior with minimal annotation cost, not learning from scratch. Furthermore, we have theoretically and experimentally proven that IPA is capable of mitigating the risk of overfitting.
>
> **ii)** Regarding the TikTok dataset evaluation, we strictly follow the evaluation protocol established by MimicMotion (ICML 2025). Most importantly, we have introduced 100 additional challenging test samples to better evaluate the model's generalization capability. Therefore, compared to prior works, our evaluation is more comprehensive.
>
> **Q3:** No human preference study.
>
> **A3:** To address your valuable feedback, we conduct a Human Preference Study (10 evaluators) on 30 challenging videos, following the MimicMotion protocol. Evaluators compare our method side-by-side against three key baselines (MimicMotion, VACE, Wan-Animate), specifically voting for "more anatomically correct, stable, and artifact-free hand structures." The summarized results below show consistent, overwhelming preference for IPA, confirming our claims.
>
> |Comparisons|Ours Win (%)|Ours Lose (%)|
> |:-|:-|:-|
> |Ours vs. MimicMotion|91.7|8.3|
> |Ours vs. VACE|87.3|12.7|
> |Ours vs. Wan-Animate|83.0|17.0|
>
> **Q4:** Is IPA really preference alignment, or just KL-regularized SFT?
>
> **A4:** For your concern, we would like to emphasize that our formulation mathematically differs from KL-regularized SFT and induces a distinct preference boundary without explicit negative samples:
>
> **i) Mathematical Distinction from Regularized SFT:** Standard KL-SFT blindly maximizes likelihood: $\mathcal{L}\_{\text{KL-SFT}} = -\mathbb{E}\_q[\log p_\theta] + \beta D\_{\text{KL}}(p_\theta | p_{\text{ref}})$. Conversely, our objective is uniquely formulated on the log-sigmoid of the KL divergence gap: $\mathcal{L}\_{\text{IPA}} = -\log \sigma \big( \beta ( D_{\text{KL}}(q | p_{\text{ref}}) - D_{\text{KL}}(q | p_\theta) ) \big)$. We have also provided the theoretical proof that IPA involves reward maximization in Sec. 4.2.
>
> **ii) How the Boundary Forms:** The sigmoid function acts as a soft margin. As $p_\theta$ successfully approaches the perfect hand structures, the loss saturates gracefully, preventing catastrophic memorization. However, if $p_\theta$ regresses towards the base model, the loss exponentially penalizes it. This establishes an asymmetric boundary.
>
> **Q5:** Validation on only one backbone.
>
> **A5:** Thanks for your suggestion. We make the clarifications below:
>
> **i) The Rationale for VACE-14B:** Our choice of VACE-14B as the primary testbed is highly deliberate. VACE is not a niche architecture; it represents the absolute state-of-the-art in massive All-in-One foundational video models.
>
> **ii) Practical Constraints During Rebuttal:** We agree that evaluating IPA on other models strengthens generalizability, but it is practically infeasible within the brief rebuttal window. Crucially, we cannot reuse the 93 VACE samples due to differing generative distributions. Applying IPA requires re-running the entire labor-intensive pipeline from scratch: generating thousands of candidates with the new model's prior, human filtering, and fine-tuning.
>
> **iii) Theoretical Generalizability:** Most importantly, given that IPA operates strictly on the mathematical probability path and spatial masking rather than relying on specific structural layers, its theoretical formulation is entirely architecture-agnostic.

---

> > ### Author Rebuttal · Reviewer_DJxR · 2026-04-03
> >
> > Partially resolved. I appreciate the additional KTO comparison and the newly added human study, which strengthen the empirical evidence. I'll raise my score. However, I remain unconvinced for the following reasons:
> > 1. The baseline issue is only partially addressed. Adding KTO is helpful, but the paper still does not compare against other highly relevant methods such as IPO, DenseDPO, or tie-aware preference optimization, which are directly related to the paper’s core motivation about ambiguous or mixed video preferences.
> > 2. The experimental scale concern remains. The method is still validated on a very limited set of manually selected samples and a small self-curated benchmark, so the evidence for generalization remains weak.
> > 3. The theoretical distinction is still not fully convincing. The rebuttal explains the objective more clearly, but I am still not fully convinced that the method learns a genuine preference boundary rather than behaving as a KL-regularized positive-only fine-tuning objective.

---

> > > ### Author Response · Authors · 2026-04-03
> > >
> > > We sincerely thank you for acknowledging our new KTO comparison and Human Preference Study, and we deeply appreciate you raising your score. We respectfully offer a final clarification on your remaining concerns:
> > >
> > > **Q1:** The paper still does not compare against IPO, DenseDPO, or Tie-aware DPO.
> > >
> > > **A1:** We respectfully reiterate that comparing IPA against these methods involves a fundamental paradigm mismatch due to their inherent data and architectural requirements.
> > >
> > > **i) Methodological Mismatch:**
> > > *   **IPO:** is fundamentally distinct from DPO and our IPA; it operates within the traditional RLHF paradigm by training a **Reward Model (RM)**. Training this RM mathematically necessitates the **strict paired winner/loser preference data**.
> > > *   **DenseDPO:** requires labor-intensive frame- or segment-level annotations to construct fine-grained pairs. This requires annotators to meticulously construct and label which specific temporal segments are "better" or "worse" in a paired video setting. **This fine-grained annotation process incurs an astronomical human labeling cost that contradicts our core objective of extreme data efficiency**.
> > > *   **Tie-aware DPO:** extends the DPO objective to accommodate "tie" scenarios, but it still fundamentally relies on the foundation of **explicitly paired data (it cannot operate exclusively on ties or single positive samples)**.
> > >
> > > **ii) Why we cannot compare:** As detailed in our paper, curating reliable good-bad pairs or dense segment rankings for complex dynamic hands is prohibitively expensive. **Because our dataset strictly consists of positive-only single samples, training RM for IPO or applying paired/cost-heavy baselines like DenseDPO/Tie-aware DPO is methodologically impossible in our setting.** KTO is implemented precisely because it is the only variant accommodating unpaired data.
> > >
> > > **iii) Explicit Positioning (Appendix B):** We anticipated this exact concern. In Appendix B.2 and B.3, we explicitly define our boundary: we do not claim IPA is inherently superior to paired-preference methods in general scenarios where strict paired data is easily obtainable. **The core contribution of IPA is providing a resource-efficient alternative for scenarios where high-quality preference pairs are scarce**.
> > >
> > > **Q2:** The experimental scale concern remains.
> > >
> > > **A2:** We respectfully clarify that our evaluation scale is neither arbitrary nor limited, but strictly aligned with domain standards, and our training scale is exactly the point of our contribution.
> > >
> > > **i) Adherence to SOTA Standards:** For the test set, **we strictly adhered to the exact evaluation protocol established by the very recent SOTA work, MimicMotion (ICML 2025).** This choice guarantees a fair, scientifically rigorous comparison against the absolute best baseline in the field. Expanding this public set arbitrarily would break comparability with established literature. In addition, we have curated 100 additional challenging test samples to better evaluate the model's generalization capability, which makes our evaluation more comprehensive.
> > >
> > > **ii) 93 Training Samples is our Core Contribution, Not a Flaw:** You mention the "very limited set of manually selected samples." However, demonstrating that we can effectively align a massive 14B foundational model using only 93 samples, without suffering from the catastrophic forgetting typical of standard SFT, is the defining scientific achievement of our IPA. It proves extreme data efficiency in a post-training regime where gathering data is hard.
> > >
> > > **Q3:** Is the method learning a preference boundary, or is it merely behaving as a KL-regularized positive-only fine-tuning objective?
> > >
> > > **A3:** To definitively prove that **IPA is not merely a regularized positive-only fine-tuning objective**, we present a new ablation study to reviewer oz83 in this rebuttal that can effectively address your concerns.
> > > Specifically, we add a standard L2 anchor-to-ref regularizer between the velocity fields of the trainable model and the reference model based on standard SFT: $\mathcal{L} = \mathcal{L}\_{\text{SFT}} + ||v_\theta - v_{\text{ref}}||^2$. All other settings remain identical to our IPA run. The results below show that while the regularizer mitigates SFT's catastrophic forgetting, its performance still vastly trails IPA.
> > >
> > > **Why IPA vastly outperforms Regularized SFT:** An explicit regularizer applies a static, merely forcing the model to "stay near the pretrained model," which prevents collapse but underfits the complex hand dynamics. Conversely, IPA's log-sigmoid formulation mathematically creates a dynamic soft-margin boundary.
> > >
> > > |Methods|FID-VID|FVD|SSIM|PSNR|
> > > |:-|:-|:-|:-|:-|
> > > |Baseline|12.5|327|0.668|18.2|
> > > |Baseline+SFT|11.8|328|0.666|16.9|
> > > |Baseline+SFT+Regularizer|11.2|308|0.673|18.5|
> > > |**IPA (Ours)**|**6.3**|**224**|**0.757**|**21.5**|
> > >
> > > We hope these further clarifications fully resolve your remaining concerns, and we would be deeply grateful if you could raise the score again.

---

### Official Review · Reviewer_ipFi · 2026-03-12

**Soundness:** 3
**Presentation:** 3
**Significance:** 3
**Originality:** 3
**Overall Recommendation:** 4
**Confidence:** 3

**Summary:**

This work proposes Implicit Preference Alignment (IPA), a method to fix the "bad hand motion" problem in human image animation. The main contribution is that, as having paired preference samples is very expensive, IPA uses only high-quality, self-generated samples to maximize an implicit reward. They also add Hand-Aware Local Optimization (HALO), which essentially masks and weighs the loss to force the model to care more about finger/hand pixels. The results show that IPA consistently outperforms state-of-the-art methods like Wan-Animate and MimicMotion, specifically reducing hand distortions in complex motion sequences.

**Compliance With Llm Reviewing Policy:**

Affirmed.

**Final Justification:**

Most of my concerns are addressed. However, the " 93 hand-picked samples " is sitll a concern. I will keep my score.

**Key Questions For Authors:**

How do you ensure the model has not simply overfit to the specific textures and motions of the 93 hand-picked samples?

**Limitations:**

Yes.

**Strengths And Weaknesses:**

## Strength

- The idea of using positive only samples from the self-generated video is an intuitive and natural idea.
- Formal formulation of IPA is sound and provides justification for the log-sigmoid loss. It shows that the objective is similar to maximizing an implicit reward while staying within a trusted KL bound.
- The HALO approach steers preference alignment towards the hand regions, via a local optimization mechanism that provides some clear benefits. in ablation studies.
- The fact that the model shows its greatest gains on the more complex hand benchmark shows the effectiveness of this method.
    - Qualitative results also show good hand movements.

## Weakness

- While the paper uses FID-VID and FVD, these metrics often fail to capture the “local hand” feasibility of the generation model. A more rigorous human preference study specifically focused on the hand would be very beneficial.
- Since the good samples are generated by the model itself,  this framework may simply be reinforcing behaviors the model already possesses. It is unclear if this approach can actually correct fundamental errors that the base model never generated correctly.
- Since only 93 hand-picked samples is used, such a small subset has a large risk of overfitting to these motions.

---

> ### Author Rebuttal · Authors · 2026-03-30
>
> ## Thanks for recognizing our contributions and promising results. Below, we respond to each Question (Q) with an Answer (A).
>
> **Q1:** A more rigorous human preference study specifically focused on the hand would be very beneficial.
>
> **A1:** For your suggestion, we make the clarifications below:
>
> **i)** Recognizing this limitation, we initially propose two mask-based, frame-wise metrics specifically designed for hand regions in our submission: SSIM-Hand and PSNR-Hand. As reported in Table 3 of our paper, our method consistently outperforms all baselines on these local metrics, providing an initial quantitative verification of our hand-aware generation capabilities.
>
> **ii)** We agree that human perception is a better standard for evaluating anatomical correctness and visual artifacts in hands. Following your valuable suggestion and the MimicMotion evaluation protocol (ICML 2025), we conduct a Human Preference Study (10 evaluators) on 30 challenging videos. For baselines, we compare our method against three representative methods: MimicMotion, VACE, and Wan-Animate. For each case, evaluators are shown the ground truth pose and video, along with the video generated by our method and a baseline video (presented side-by-side in randomized order). Evaluators are asked to vote for the video that exhibited "more anatomically correct, stable, and artifact-free hand structures," with options for "Win" and "Lose". The summarized results below show consistent, overwhelming preference for IPA, confirming our claims.
>
> |Comparisons|Ours Win (%)|Ours Lose (%)|
> |:-|:-|:-|
> |Ours vs. MimicMotion|91.7|8.3|
> |Ours vs. VACE|87.3|12.7|
> |Ours vs. Wan-Animate|83.0|17.0|
>
> **Q2:** Since the good samples are generated by the model itself, this framework may simply be reinforcing behaviors the model already possesses. It is unclear if this approach can actually correct fundamental errors that the base model never generated correctly.
>
> **A2:** To directly answer your question: We completely agree that if a base model has zero prior knowledge of hand structures (i.e., it never generates them correctly under any circumstances), IPA alone cannot synthesize this knowledge from thin air. Here is why our IPA effectively corrects hand generation errors in this specific context:
>
> **i)** Large-scale video generators possess immense prior knowledge of human anatomy due to their massive pre-training data. The fundamental issue is not that the base model cannot generate correct hands; rather, it is that during highly complex and dynamic conditioning, the model's distribution becomes unstable, leading to collapsed or distorted hands in the vast majority of cases. 93 high-quality samples explicitly prove that the base model does possess the correct behaviors, but this knowledge is highly suppressed in its latent distribution.
>
> **ii)** IPA as a probability mass shift, not just memorization. Our framework is not merely "reinforcing what the model already does" in a naive sense. Instead, IPA mathematically acts as a distribution alignment mechanism. By maximizing the implicit reward function, IPA shifts the probability mass of the model's generation trajectory. It takes the rare, high-fidelity modes and pulls the policy distribution towards them, making these rare, correct structural behaviors the dominant, high-probability outputs.
>
> **Q3:** Since only 93 hand-picked samples is used, such a small subset has a large risk of overfitting to these motions.
>
> **A3:** For your concern, we make the clarifications below:
>
> **i) Theoretical Prevention of Overfitting via KL-Divergence Constraint.** Unlike standard Supervised Fine-Tuning (SFT), which blindly maximizes the likelihood of the training data, our IPA objective is fundamentally grounded in constrained reward maximization. As theoretically derived in Section 4.2 (Objective 2) and Eq. 11, our loss function explicitly incorporates a crucial hyperparameter, which acts as a stringent regularizer. controls the strength of the KL-Divergence penalty between our preference-aligned policy and the pre-trained reference model.
>
> **ii) Empirical Evidence of Strong Generalization.** As explicitly stated in Section 5.1, our evaluations are conducted on the standard TikTok benchmark and our newly proposed challenging benchmark. Crucially, all test samples are strictly disjoint from our 93 training samples. The motions and character appearances in the test sets are entirely new to the model. If the model has overfitted, its performance on these novel test sets would have plummeted. Instead, as shown in Tabs. 1 and 2, our IPA achieves state-of-the-art results.
>
> **iii)** To further illustrate the power of IPA against overfitting, we provide a direct ablation study in Appendix A.1 (Table 6). When we train the exact same model on the exact same 93 samples using standard SFT, the model indeed suffers from severe mode collapse. Conversely, our IPA framework significantly improves performance.

---

> > ### Author Rebuttal · Reviewer_ipFi · 2026-04-06
> >
> > Most of my concerns are addressed. However, the " 93 hand-picked samples " is sitll a concern. I will keep my score.

---

> > > ### Author Response · Authors · 2026-04-06
> > >
> > > We sincerely thank you for reviewing our rebuttal and for recognizing that most of your concerns have been successfully addressed. We are truly grateful for your continued support and for maintaining your positive score.
> > >
> > > Regarding your concern about the "93 hand-picked samples," we understand your intuition. However, we respectfully ask you to view this number through the specific lens of our core claim: **developing a data-efficient post-training framework.**
> > >
> > > **i) Post-Training Alignment vs. Large-Scale Pre-Training:** In traditional supervised learning or massive pre-training, 93 samples would indeed be insufficient. However, our 14B-parameter foundation model already possesses extensive prior knowledge of human anatomy. Our objective is not to teach the model from scratch, but to perform preference alignment, steering its existing capabilities toward human-preferred, high-fidelity structures. **In this post-training regime, utilizing a minimal set of exceptionally high-quality data to align a powerful prior is exactly the central goal and scientific contribution of our work.**
> > >
> > > **ii) Theoretical and Empirical Prevention of Overfitting:** The primary risk of relying on a small dataset is overfitting. We explicitly designed IPA to mitigate this:
> > > *   **Theoretically:** Our objective (Eq. 11) is mathematically formulated with a $\beta$-KL constraint that creates a dynamic soft-margin, rigorously preventing the model from collapsing into memorizing the 93 samples.
> > > *   **Empirically:** As demonstrated in our experiments, standard SFT on these identical 93 samples suffers from catastrophic forgetting (Appendix A.1). In stark contrast, IPA successfully achieves state-of-the-art performance on entirely unseen, highly complex test sets. This conclusively proves that the model generalized the intrinsic geometric rules of hands rather than overfitting to the training data.
> > >
> > > We hope these clarifications fully resolve your final concerns regarding the sample size. Thank you once again for your rigorous review and for your continued positive evaluation of our work!

---

### Decision · Program_Chairs · 2026-04-30

**Decision:**

Accept (regular)

**Comment:**

The paper proposes Implicit Preference Alignment (IPA), a data-efficient post-training method that improves hand generation quality in human image animation using only positive (good) samples, without requiring paired preference data. The method combines a log-sigmoid loss on the KL divergence gap with Hand-Aware Local Optimization (HALO) for spatial targeting of hand regions.

Strengths:
- Well-motivated practical problem: hand quality in video generation is a real bottleneck, and constructing strict preference pairs for dynamic hand regions is genuinely impractical
- Strong empirical results: substantial improvements over baselines (FID-VID 5.9 vs 8.6, SSIM-Hand 0.606 vs 0.544) with only 93 curated good samples and 1000 LoRA training steps
- Human preference study added in rebuttal (83-92% win rates) validates perceptual quality
- The SFT+L2 ablation effectively demonstrates that IPA's gains come from the sigmoid mechanism, not just proximity to the pretrained model
- HALO is simple and effective, validated by ablation

Weaknesses:
- The theoretical framing overstates the contribution. As Reviewer oz83 identified and the authors acknowledged, IPA's loss (Eq. 27) is structurally equivalent to Flow-DPO with the negative sample term removed. The novelty is the first-principles derivation and practical justification for this reduction, not a new loss function.
- The beta characterization as a "KL constraint" is imprecise. The evidence is more consistent with beta controlling sigmoid saturation speed (how quickly gradients vanish during training). The beta ablation (Table 7) showing symmetric degradation around beta=600 supports a saturation-speed interpretation over a KL-constraint interpretation.
- Reviewer DJxR's concern about the theoretical distinction from KL-regularized SFT is valid. The SFT+L2 ablation shows IPA is empirically superior, but the mechanism explanation (sigmoid saturation as adaptive early stopping vs implicit KL constraint) remains ambiguous.
- Evaluation is on a single backbone (VACE-14B) with a small training set (93 samples), following established protocols but limiting generalizability evidence.

Recommendation: Accept. Three of four reviewers support acceptance.

Revision requirements for camera-ready:
- State in the main text that IPA's loss is structurally equivalent to positive-only Flow-DPO, while the derivation provides independent theoretical justification (per oz83's condition)
- Include the Delta training curve showing sigmoid saturation behavior (per oz83's condition)
- Revise the beta characterization to present both the KL-constraint and saturation-speed interpretations, acknowledging that the mechanism is not definitively resolved
- Include the SFT+L2 ablation, KTO comparison, and human preference study results from the rebuttal